# Citrullination of pyruvate kinase M2 by PADI1 and PADI3 regulates glycolysis and cancer cell proliferation

Sébastien Coassolo[1,2,3,4,6], Guillaume Davidson [1,2,3,4], Luc Negroni[1,2,3,4], Giovanni Gambi[1,2,3,4], Sylvain Daujat[1,2,3,4,5], Christophe Romier [1,2,3,4] & Irwin Davidson [1,2,3,4✉]

Chromodomain helicase DNA binding protein 4 (CHD4) is an ATPase subunit of the Nucleosome Remodelling and Deacetylation (NuRD) complex that regulates gene expression. CHD4 is essential for growth of multiple patient derived melanoma xenografts and for breast cancer. Here we show that CHD4 regulates expression of PADI1 (Protein Arginine Deiminase 1) and PADI3 in multiple cancer cell types modulating citrullination of arginine residues of the allosterically-regulated glycolytic enzyme pyruvate kinase M2 (PKM2). Citrullination of PKM2 R106 reprogrammes cross-talk between PKM2 ligands lowering its sensitivity to the inhibitors Tryptophan, Alanine and Phenylalanine and promoting activation by Serine. Citrullination thus bypasses normal physiological regulation by low Serine levels to promote excessive glycolysis and reduced cell proliferation. We further show that PADI1 and PADI3 expression is up-regulated by hypoxia where PKM2 citrullination contributes to increased glycolysis. We provide insight as to how conversion of arginines to citrulline impacts key interactions within PKM2 that act in concert to reprogramme its activity as an additional mechanism regulating this important enzyme.

---

[1] Institut de Génétique et de Biologie Moléculaire et Cellulaire, Equipe Labélise Ligue Contre le Cancer, Illkirch, France. [2] Centre National de la Recherche Scientifique, Paris, France. [3] Institut National de la Santé et de la Recherche Médicale, Paris, France. [4] Université de Strasbourg, Strasbourg, France. [5] Biotechnology and Cell Signaling, CNRS UMR7242, 300 Bd Sébastien Brandt, Illkirch, France. [6] Present address: Discovery Oncology, Genentech, South San Francisco, CA, USA. ✉email: irwin@igbmc.fr

A hallmark of cancer cells is the high glycolysis and lactic acid production under aerobic conditions, a metabolic state known as the Warburg effect[1]. Tumour tissues accumulate increased amounts of glucose used not only for energy production, but also for anabolic reactions. Glycolytic intermediates are notably used for de novo synthesis of nucleotides or amino acids like glycine and serine produced in large amounts to sustain high rates of cancer cell proliferation[2,3]. Coupling of energy production via glycolysis to the availability of the intermediates required for nucleotide and amino acid synthesis is controlled in large part by an alternatively spliced isoform of the enzyme pyruvate kinase called PKM2 expressed in proliferating embryonic and cancer cells[4,5]. Unlike the PKM1 isoform that is constitutively active, PKM2 activity is positively regulated by serine (Ser), fructose 1,6-biphosphate (FBP) an intermediate of the glycolytic pathway and succinylaminoimidazole-carboxamide riboside (SAICAR), an intermediate in de novo purine nucleotide synthesis[4,6,7]. High levels of these molecules stimulate PKM2, but when their levels are lowered through excessive glycolysis, PKM2 activity is inhibited by amino acids such as tryptophan (Trp), alanine (Ala) and phenylalanine (Phe) that compete with Ser to allosterically regulate PKM2 activity[8–10]. Through this complex feedback loop, PKM2 couples glycolytic flux to the level of critical intermediate metabolites. PKM2 activity is also regulated by post-translational modifications, such as tyrosine phosphorylation or proline hydroxylation under hypoxia[11,12].

Melanoma cells are no exception to the Warburg effect, showing high levels of aerobic glycolysis induced by transformation with oncogenic BRAF or NRAS[13]. Treatment with the oncogenic BRAF inhibitor vemurafenib down-regulates aerobic glycolysis that is regained in resistant cells[14]. Transcription factor MITF (Microphthalmia associated transcription factor) regulates many parameters of melanoma cell physiology including metabolism[15]. MITF directly regulates PPARGC1 and cells with high MITF expression show higher oxidative phosphorylation compared to cells with low MITF[16,17].

Bossi et al[18] performed an shRNA dropout screen to identify proteins involved in epigenetics and chromatin functions essential for patient derived melanoma xenograft (PDX) growth. This screen identified the ATPase subunit of the PBAF chromatin remodelling complex BRG1 along with CHD4, the catalytic ATPase subunits of the Nucleosome Remodelling and Deacetylation (NuRD) complex, as essential for tumour formation by all tested melanoma PDX. NuRD, is an epigenetic regulator of gene expression, acting in many, but not all[19], contexts as a co-repressor that remodels chromatin through its ATPase subunits and deacetylates nucleosomes through its HDAC1 and HDAC2 subunits[20–24]. CHD4 has also been reported to be essential in breast cancer[25]. Here, we describe a pathway where CHD4 regulates expression of the PADI1 (Protein Arginine Deiminase 1) and PADI3 enzymes (hereafter designated as PAD1 and PAD3) that convert peptidyl-arginine to citrulline[26]. Increased PAD1 and PAD3 expression enhances citrullination of the key glycolytic enzyme PKM2 leading to excessive glycolysis, lowered ATP levels and slowed cell growth. In human cancers, PADI1 and PADI3 expression is regulated by hypoxia stimulating PKM2 activity and contributing to the increased glycolysis seen under hypoxic conditions.

## Results
**CHD4 regulates PAD1 and PAD3 expression in melanoma cells.** We performed siRNA CHD4 silencing in a collection of melanoma cells. Silencing was specific for CHD4 and did not affect CHD3 expression as measured by RT-qPCR and confirmed by immunoblot (Fig. 1a, b). CHD4 silencing also did not

appreciably affect expression of the lineage-specific transcription factors MITF or SOX10. In agreement with the results of the previous shRNA dropout screen, siRNA-mediated CHD4 silencing reduced colony forming capacity, increased the proportion of slow proliferating cells (Fig. 1a–d, and Supplementary Fig. 1), but did not induce apoptosis (Fig. 1e). Analogous results were seen were seen following CHD3 silencing. CHD4 silencing had a less dramatic effect than silencing of MITF that induces a potent cell cycle arrest and senescence[27,28]. RNA-seq following CHD4 silencing in melanoma cells identified more than 1000 up-regulated genes compared to 364 down-regulated genes showing that CHD4 acted primarily as a transcriptional repressor (Fig. 1f and Supplementary Dataset 1). Up-regulated genes were enriched in several signalling pathways (Fig. 1g).

De-regulated expression of selected genes upon CHD4 silencing was confirmed by RT-qPCR on independent RNA samples in 501Mel, MM117 and SK-Mel-28 melanoma cells (Fig. 1h–j). While RAC2 and CNTFR showed comparable up-regulation in the 3 lines, many other genes were differentially regulated. For example, THY1 was potently induced in 501Mel cells, but it expression was mildly down-regulated in Sk-Mel-28 cells, whereas MAP2 that was down-regulated in 501Mel cells was up-regulated in MM117 cells. In contrast, expression of PADI1 and PADI3 was strongly induced in all tested melanoma lines (Fig. 2a, b). PADI3 expression was almost undetectable and potently activated by CHD4 silencing, whereas some lines had low basal PADI1 levels also strongly stimulated by CHD4 silencing. RNA-seq further showed that expression of PADI2 and PADI4 was low to undetectable and their expression was not induced by siCHD4 silencing (Supplementary Dataset 1).

The PADI1 and PADI3 genes are located next to each other (Supplementary Fig. 2). ChIP-seq in melanoma cells revealed that CHD4 occupied an intronic regulatory element in PADI1 immediately adjacent to sites occupied by transcription factors CTCF and FOSL2 (AP1). This element is predicted to regulate both the PADI1 and PADI3 genes (Supplementary Fig. 2) and is further marked by H2AZ, H3K4me1, BRG1 and ATAC-seq, but not by MITF and SOX10.

Analyses of the Cancer Cell Line Encyclopaedia (CCLE) showed a strong correlation of PADI1 and PADI3 expression indicating their co-regulation was not restricted to melanoma cells (Supplementary Fig. 3). Remarkably, expression of both PADI1 and PADI3 was negatively correlated with CHD4 (Supplementary Fig. 2b-c). These data support the idea that CHD4 repressed PADI1 and PADI3 in many cancer cell lines (see also below).

**PAD1 and PAD3 citrullinate glycolytic enzymes and stimulate glycolysis.** To identify potential PAD1 and PAD3 substrates in melanoma cells, we made protein extracts from siC and siCHD4 cells, performed immunoprecipitation (IP) with a pan-citrulline antibody and analysed precipitated proteins by mass-spectrometry (Fig. 2c and Supplementary Dataset 2). A statistical analysis of the data using the Perseus software revealed an increased number of total peptide spectral matches (PSMs) and PSMs for citrullinated peptides following CHD4 silencing. A set of 520 proteins was enriched in the pan-citrulline IP from siCHD4 cells (Fig. 2c and Supplementary Dataset 2). Ontology analyses showed strong enrichment in proteins involved in translation, proteasome, spliceosome and multiple aspects of metabolism, including glycolysis and the pentose phosphate pathway (Supplementary Fig. 4a). Comparison with public datasets of citrullinated human proteins indicated that around half of the citrullinated proteins identified in rheumatoid arthritis were present in our data set while around 30% of the proteins

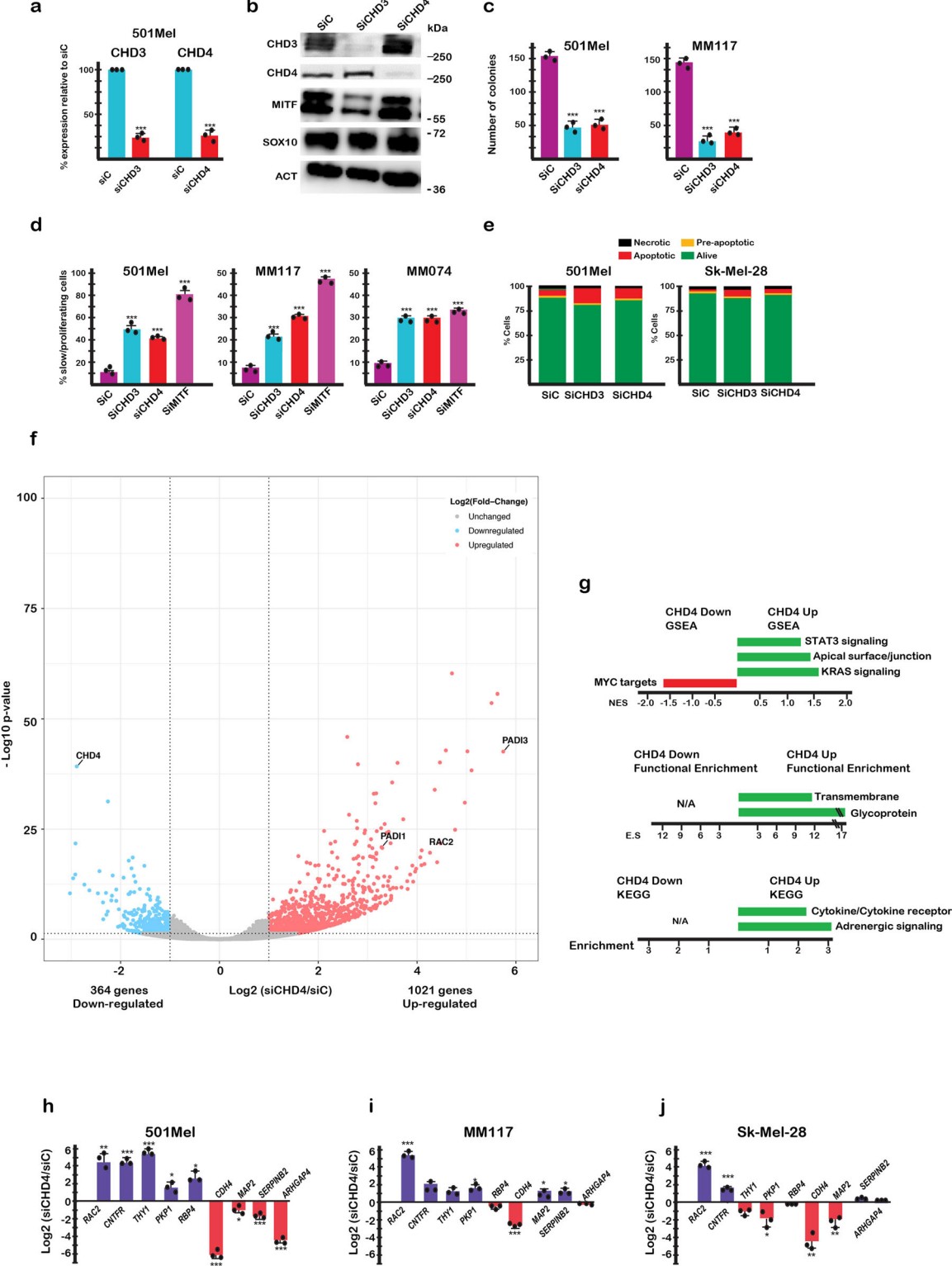

**Fig. 1 CHD3 and CHD4 are required for normal melanoma cell proliferation. a**, **b** 501Mel cells were transfected with the indicated siRNAs and CHD3 and CHD4 expression evaluated by RT-qPCR or by immunoblot along with that of MITF and SOX10. **c** The indicated cell lines were transfected with siRNA and after reseeding the number of colonies counted after 10 days. **d** The indicated cell lines were transfected with siRNAs and cell proliferation evaluated by cell trace violet assay. **e** The indicated cell lines were transfected with siRNA and apoptosis detected by FACs after labelling with Annexin-V. Silencing of MITF known to induce cell cycle arrest and senescence was included as a control. **f** Volcano plot showing changes in gene expression following CHD4 silencing. Genes up or down-regulated based on Log2 fold-change >1/<-1 with an adjusted $p$-value <0.05 were identified. **g** Ontology analyses of CHD4 regulated genes. Shown are the enrichment scores for GSEA, as well as David functional enrichment and KEGG pathway categories. **h–j** Verification of deregulated expression of selected genes following siCHD4 in independent RNA samples from 501Mel, MM117 and Sk-Mel-28 cells. In all experiments $n = 3$ biological replicates and unpaired $t$-test with two tailed $p$-value analyses and confidence interval 95% were performed by Prism 5. $p$-Values: $*p < 0.05$; $**p < 0.01$; $***p < 0.001$. Data are mean ± SEM.

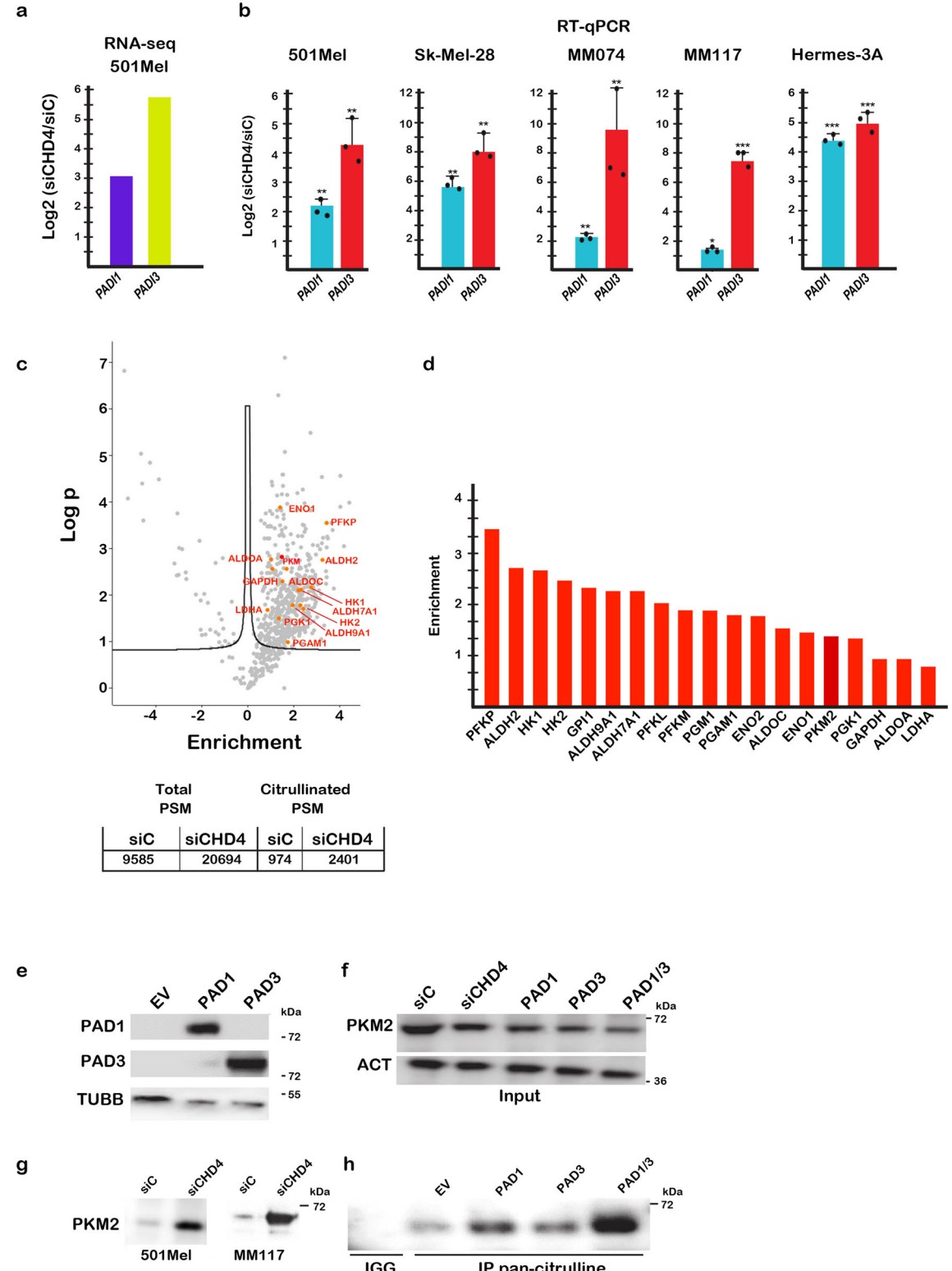

found here were previously found in human tissues[29,30] (Supplementary Fig. 4b, c).

Glycolysis was amongst the pathways strongly enriched in siCHD4 cells with enhanced citrullination of multiple enzymes (Fig. 2d). We focussed on PKM2, a highly regulated enzyme playing a central role in integrating control of glycolysis with cellular metabolic status and cell cycle[31]. PKM2 converts phosphoenolpyruvate (PEP) to pyruvate then converted to lactic acid. To investigate PKM2 citrullination, melanoma cells were transfected with siC, siCHD4 or vectors allowing ectopic expression of PAD1 and PAD3 (Fig. 2e). While CHD4 silencing or PAD1/3 expression did not alter overall PKM2 levels (Fig. 2f), strongly increased amounts of PKM2 were detected in the pan-citrulline IP following siCHD4 compared to siC in both 501Mel and MM117 melanoma cells and after ectopic PAD1 and PAD3

**Fig. 2 CHD4 regulates *PADI1* and *PADI3* expression and citrullination of their substrates. a, b** *PADI1* and *PADI3* expression in the indicated cells lines following CHD4 silencing shown by RNA-seq (**a**) and RT-qPCR (**b**). $n = 3$ biological replicates. and unpaired *t*-test with two tailed *p*-value analyses and confidence interval 95% were performed by Prism 5. *p*-Values: *$p < 0.05$; **$p < 0.01$; ***$p < 0.001$. Data are mean ± SEM. **c** Volcano plot showing proteins with increased or decreased total PSMs after immunoprecipitation with pan-citrulline antibody. Increases in number of total and citrullinated PSMs in CHD4 silenced cells is shown below the plot. **d** Enrichment for a collection of glycolytic enzymes following CHD4 silencing. **e** Immunoblot showing expression of recombinant PAD1 and PAD3 in cells transfected with the corresponding expression vectors of the empty vector (EV). **f** Immunoblot showing expression of PKM2 in cells after CHD4 silencing or transfection with the PAD1 and PAD3 vectors in the cell extracts used for immunoprecipitation with pan-citrulline antibody. **g** PKM2 in the pan-citrulline IPs from 501Mel or MM117 cells. **h** Immunoblot showing PKM2 in the pan-citrulline IP after transfection with the PADI1 and/or PADI3 expression vectors.

expression, particularly upon co-expression of both enzymes (Fig. 2g, h).

Previous studies in human or mouse cells[29,32] identified at least 3 PKM2 arginine residues that were subject to citrullination R106, R246 and R279 (Supplementary Fig. 5a). The above mass-spectrometry detected R106 and R279 citrullination, whereas that of R246 could not be unambiguously determined as it was located the C-terminus of the peptide (Supplementary Fig. 5b–d). A similar situation was also seen for R489 that had not been found in previous studies (Supplementary Fig. 5e). To confirm their citrullination, we generated antibodies against synthetic peptides corresponding to citrullinated R106 and R246, residues predicted to play a role in regulating PKM2 activity (see below). In dot-blot assays, each of these antibodies showed strong signal for the citrullinated peptide, but little for the equivalent wild-type peptide with arginine (Supplementary Fig. 6a–d). Similarly, citrullinated R106 antibody did not recognise citrullinated R246 and vice-versa, and signal for each antibody was lost after siPKM2 silencing (Supplementary Fig. 6e). Immunoblots on extracts from cells transfected with siCHD4 or PAD1 and PAD3 expression vectors showed enhanced signal for PKM2 compared to the control transfected cells indicating increased citrullination of these two arginine residues (Supplementary Fig. 6g, h). The R489 peptide was too hydrophobic to obtain soluble peptide for antibody production and thus its citrullination could not be confirmed.

To determine if siCHD4 silencing and enhanced PKM2 citrullination altered glycolysis, we profiled melanoma cell metabolism in real time. CHD4 silencing increased the basal OCR (oxygen consumption rate) and ECAR (extracellular acidification rate), markedly increased maximum OCR and ECAR and decreased the OCR/ECAR ratio due to the increased ECAR values (Fig. 3a–d). ECAR was blocked using 2-deoxy-D-glucose confirming that it was due to increased glycolysis (Fig. 3c). Increased glycolysis and lactic acid production diverts pyruvate from oxidative metabolism a more efficient ATP source. Consequently, excessive glycolysis following CHD4 silencing led to decreased intracellular ATP levels (Fig. 3e). Similarly, CHD4 silencing increased PKM2 activity that mirrored the increased glycolysis of the living cells (Fig. 3f and Supplementary Fig. S7a, b). In contrast, no increased PKM activity was seen upon siCHD4 silencing in primary WI-38 fibroblasts (Supplementary Fig. 7a, b) that expressed PKM1 and not PKM2 (Supplementary Fig. 6h) and no increase in glycolysis was seen upon ectopic PAD1 and PAD3 expression (Supplementary Fig. 7d).

The increased glycolysis seen upon CHD4 silencing was strongly diminished when PAD1 and PAD3 were additionally silenced (Fig. 3g). In contrast, exogenous expression of PAD1, PAD3 or both stimulated glycolysis (Fig. 3h). Consistent with increased glycolysis, PAD1/3 expression led to reduced intracel-lular ATP levels (Fig. 3i) and reduced cell proliferation (Fig. 3j). PAD1 and PAD3 were therefore necessary and sufficient for increased glycolysis accounting for the effect seen upon CHD4 silencing. Similar to CHD4 silencing, expression of

PAD1, PAD3 or both stimulated PKM2 enzymatic activity (Fig. 3f and Supplementary Fig. 7a–c). Increased glycolysis after siCHD4 or PAD1/3 expression could therefore be attributed to increased PKM2 activity in keeping with its previously described role as the key regulatory enzyme.

It has previously been shown that treatment of melanoma cells with BRAF inhibitors induces metabolic reprogramming, strongly reducing glycolysis[14]. Moreover, dependence on glycolysis sensitizes melanoma cells to the effects of BRAF inhibition[33]. Consistent with these observations, CHD4 silencing or ectopic PAD1/3 expression that increased glycolysis sensitised Sk-Mel-28 cells to the effects of the BRAF inhibitor vemurafenib (Supplementary Fig. 8). Hence, by regulating glycolysis, CHD4 si-lencing or PAD1 and PAD3 expression acts to modulate melanoma cell sensitivity to BRAF inhibition. Nevertheless, the effect of CHD4 silencing had more potent effects on vemurafenib sensitivity that ectopic PAD1 and PAD3 expression suggesting additional pathways are affected.

**PAD1 and PAD3 stimulate glycolysis in a variety of cancer cell types**. As mentioned above, *PADI1* and *PADI3* were co-ordinately regulated in multiple types of cancer cells and their expression was inversely related to that of CHD4. To test this idea, we silenced CHD4 in a variety of cancer cell lines. CHD4 silencing in SiHa cervical carcinoma cells strongly diminished their colony forming capacity (Fig. 4a), potently increased *PADI3* expression (Fig. 4b) and stimulated glycolysis (Fig. 4c, d). Moreover, glycolysis was stimulated by ectopic PAD1/3 expression leading to reduced OCR/ECAR ratio and ATP levels (Fig. 4e–g). In HeLa cells, CHD4 silencing reduced colony forming capacity and activated *PADI1* and *PADI3* expression (Fig. 4h, i). Glycolysis was stimulated by both CHD4 silencing and ectopic PAD1/3 expression (Fig. 4j). CHD4 silencing strongly stimulated *PADI1* and *PADI3* expression in MCF7 breast cancer cells and increased glycolysis (Fig. 4k, l). Analogous results were observed in two different types of renal cell carcinoma cell lines, UOK-109 translocation renal cell carcinoma cells (Fig. 4m–q) and A498 clear cell renal carcinoma cells (Fig. 4r–v). Therefore, in cell lines from distinct cancer types, CHD4 silencing or ectopic PAD1/3 expression increased glycolysis and negatively impacted cell proliferation.

**Citrullination reprogrammes PKM2 allosteric regulation**. As described in the introduction, PKM2 isoform activity is positively regulated by Ser and FBP and negatively by Trp, Ala and Phe, thus coupling glycolytic flux to the level of critical intermediate metabolites[4–6]. PKM2 allosteric regulation involves three distinct enzyme conformations[8,9,34] (Supplementary Fig. 9a). In the apo (resting) state, in absence of small molecules and ions, the PKM2 N-terminal and A domains adopt an active conformation, but the B domain is in an inactive conformation. In the activated R-state, binding of FBP or Ser and magnesium, stabilises the N and A domains in their active conformation, and rotates the B domain

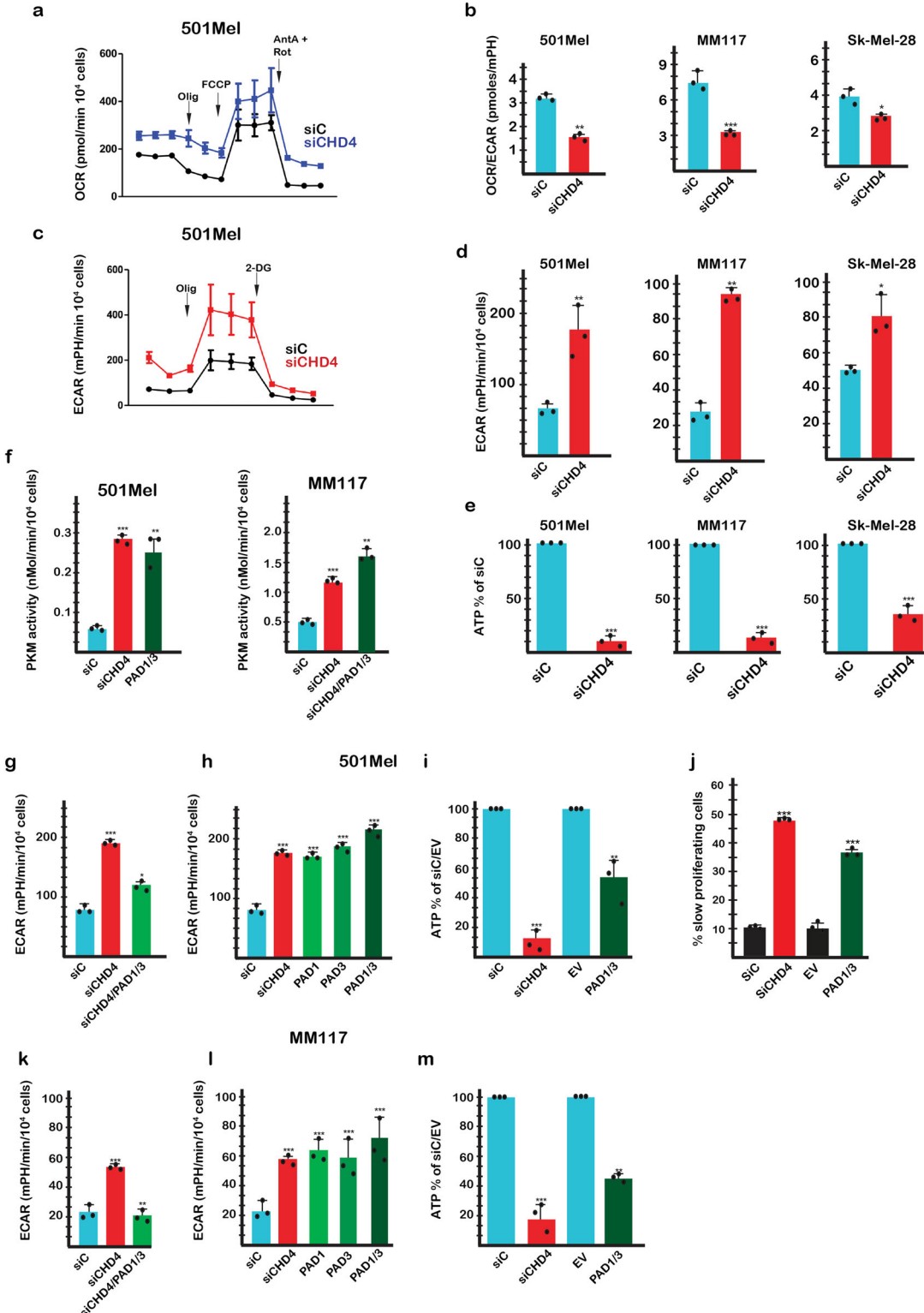

**Fig. 3 CHD4 silencing regulates glycolysis and cell proliferation. a** Effect of CHD4 silencing on basal and maximal OCR values in 501Mel cells. **b** Effect of CHD4 silencing on the basal OCR/ECAR ratio in the indicated cell types. **c, d** Effect of CHD4 silencing on basal and maximal ECAR values in 501Mel cells and basal ECAR values in the indicated cell types. **e** CHD4 silencing reduces intracellular ATP levels in the indicated cell lines. **f** Stimulation of PKM2 enzymatic activity in extracts from cells under the indicated conditions. **g, h** ECAR values in 501Mel cells following transfection with indicated siRNAs or expression vectors. **i** Intracellular ATP levels following CHD4 silencing or PAD1/3 expression. EV = empty expression vector control. **j** Reduced cell proliferation following PAD1/3 expression. **k–m** ECAR values and intracellular ATP levels in MM117 cells following transfection with indicated siRNAs or expression vectors. ECAR and OCR and ATP values were determined from $n = 3$ biological replicates with 6 technical replicates for each N in the case of OCR and ECAR. $n = 3$ biological replicates and unpaired $t$-test with two tailed $p$-value analyses and confidence interval 95% were performed by Prism 5. $p$-Values: *$p < 0.05$; **$p < 0.01$; ***$p < 0.001$. Data are mean ± SEM. Values for PKM2 enzymatic activity were determined by Prism 5 using a 2-way ANOVA test.

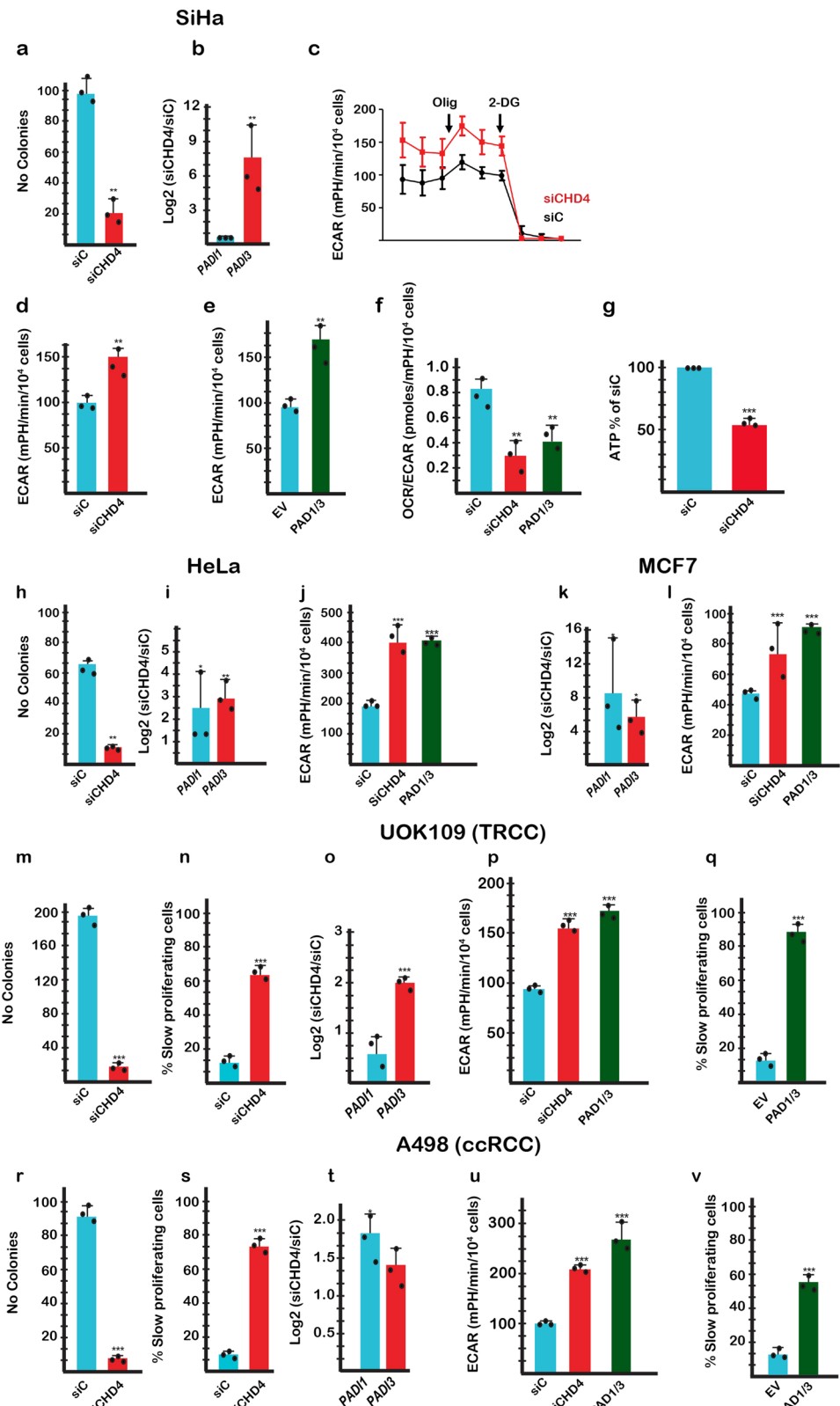

towards the A domain that together form the active site. In the inactive T-state, upon binding of inhibitory amino acids (Trp, Ala and Phe), the B domain adopts a partially active conformation, but the N and A domains undergo structural changes and disorganise the active site. The structural changes observed between the different PKM2 states are reinforced allosterically by organisation into a tetramer that is essential for enzyme function.

In previous studies and as described here PKM2 can be citrullinated at 3 arginine residues R106, R246, R279, and potentially also at R489. Citrullination of R106 and R246 was confirmed by immunoblot (Supplementary Fig. 6f, g). In the apo state, R246 forms hydrogen bonds between its guanidino group and the main chain carbonyls of V215 and L217 at the pivotal point where the B domain moves between its active and inactive

**Fig. 4 Citrullination regulates glycolysis and proliferation in multiple types of cancer cells. a** Diminished colony forming capacity of SiHa cells following CHD4 silencing. **b** *PADI1* and *PADI3* expression in SiHa cells following CHD4 silencing. **c, d** Basal and maximal glycolysis in SiHa cells following CHD4 silencing. **e** Glycolysis in SiHa cells following PAD1/3 expression. **f** OCR/ECAR ratio in SiHa cells following CHD4 silencing or PAD1/3 expression. **g** Intracellular ATP levels in SiHa cells following CHD4 silencing. **h–j** Colony forming capacity, *PADI1*, *PADI3* expression and glycolysis in HeLa cells following CHD4 silencing or PAD1/3 expression as indicated. **k, l** *PADI1*, *PADI3* expression and glycolysis in MCF7 cells following CHD4 silencing or PAD1/3 expression as indicated. **m–q** Colony forming capacity, *PADI1*, *PADI3* expression and glycolysis and proliferation in UOK-109 translocation renal cell carcinoma cells following CHD4 silencing or PADI1/3 expression as indicated. **r–v** Colony forming capacity, *PADI1*, *PADI3* expression and glycolysis and proliferation of A498 clear cell renal carcinoma cells following CHD4 silencing or PAD1/3 expression as indicated. For RT-qPCR, ATP levels and clonogenicity, $n = 3$ biological replicates For ECAR: $n = 3$ biological replicates with 6 technical replicates for each N. Unpaired $t$-test with two tailed $p$-value analyses and confidence interval 95% were performed by Prism 5. $p$-Values: *$p < 0.05$; **$p < 0.01$; ***$p < 0.001$. Data are mean ± SEM.

conformations[34] (Supplementary Fig. 9b). This interaction contributes to maintaining the inactive B domain conformation in the apo state and is lost in the R- and T-states. R246 citrullination should weaken interaction with V215 and L217 facilitating release of the B domain from its inactive conformation. As no consequence of R279 citrullination on PKM2 activity could be readily predicted, this residue was not further studied.

R106 participates in the free amino acid binding pocket. In the apo state, R106 mostly faces the solvent, but upon free amino acid binding, it rotates towards the pocket where its guanidino group interacts with the carboxylate group of the bound amino acid and the P471 main chain carbonyl[6,8,9] (Fig. 5a). Ser forms a hydrogen bond network with the N and A domains stabilising their active conformations, whereas the hydrophobic side chains of Trp, Ala, or Phe cause displacement of the N-domain outwards leading to the allosteric changes that characterise the inactive T-state (Fig. 5a).

Transition between the R- and T-states is finely regulated by changes in the relative concentrations of Ser versus Trp, Ala and Phe that compete for binding to the pocket[9]. Loss of R106 positive side chain charge upon citrullination will diminish its ability to interact with the carboxylate group of the free amino acids. Due to its extended network of hydrogen bonds within the pocket and as it does not modify the active conformations of the N and A domains, it is possible that binding of Ser is less affected than that of the hydrophobic amino acids that induce important structural changes within the N and A domains. Consequently, R106 citrullination could reduce the inhibitory effect of Trp, Ala and Phe thereby shifting the equilibrium towards activation by Ser.

To test the above hypotheses, we asked if citrullination modulated glycolysis under different conditions. When cells were grown in absence of Ser, basal glycolysis was reduced and was no longer stimulated upon siCHD4 or PAD1/3 expression (Fig. 5b). On the other hand, exogenous Ser stimulated basal glycolysis to a level that was not further increased by siCHD4 (Fig. 5c). In contrast, basal glycolysis was reduced by exogenous Trp, but remained stimulated by siCHD4 and by PAD1/3 expression (Fig. 5d). Similarly, glycolysis was stimulated by siCHD4 in presence of increasing Phe concentrations (Fig. 5e), an effect particularly visible in MM117 cells where despite strongly inhibited basal glycolysis, stimulation was seen upon siCHD4 (Fig. 5f). PAD1/3 expression also stimulated glycolysis in presence of exogenous Ala (Fig. 5g). PKM2 citrullination did not therefore bypass the requirement for Ser, while excess Ser mimicked stimulation seen by siCHD4. In contrast, siCHD4 or PAD1/3 expression diminished inhibition by Trp/Ala/Phe, consistent with the idea that R106 citrullination modified the equilibrium in favour of the activator Ser. The inhibitory effects of Phe and Trp on glycolysis were not seen in WI-38 fibroblasts that expressed PKM1 and not PKM2 consistent with previous results showing that PKM1 is not allosterically regulated by these ligands and hence that the effects were principally mediated via PKM2 (Supplementary Fig. 6d).

The central role of PKM2 was further supported by analysing PKM2 enzymatic activity. Addition of exogenous Phe or Trp strongly inhibited PKM2 activity in control cell extracts, whereas this inhibition was partially overcome in extracts from siCHD4 or PAD1/3 expressing cells (Fig. 5h, i and Supplementary Fig. 7e–g). At higher inhibitor concentrations, PKM2 activity was most efficiently restored by PAD1/3 overexpression. Thus, citrullination attenuated PKM2 inhibition by Phe and Trp to restore both its enzymatic activity and glycolysis in living cells.

R489 is directly involved in FBP binding with strong interactions between its guanidino group and the FBP 1' phosphate group (Fig. 6a). Despite its extensive interaction network with PKM2, FBP binding is severely reduced upon mutation of R489 into alanine[8,10]. Hydrogen bonding with R489 therefore plays a critical role in FBP binding that should be diminished by loss of its side chain charge if indeed this residue is subject to citrullination, hence suggesting that activation of PKM2 by citrullination required weakening of its interaction with FBP.

In agreement with this idea, increasing concentrations of exogenous FBP had little effect on basal glycolysis, but blocked stimulation by siCHD4 (Fig. 6b). Addition of exogenous Ser at low FBP concentration (0.5 mM) augmented basal glycolysis and re-established stimulation by siCHD4. In contrast, at higher FBP concentration (2.0 mM), little increase in basal or siCHD4-stimulated glycolysis was seen in presence of exogenous Ser. Increasing FBP therefore inhibited Ser and citrullination-dependent stimulation of glycolysis.

At low concentrations of exogenous FBP, increasing concentrations of exogenous Phe lowered basal glycolysis, whereas higher FBP concentrations overcame the Phe-induced repression (Fig. 6c, d), consistent with the known antagonistic effect of these ligands[9,10]. At low FBP concentrations and in presence of Phe, siCHD4 stimulated glycolysis, but to a lower level than seen in absence of FBP, whereas higher FBP concentrations blocked stimulation.

These observations indicated that increased FBP inhibited the ability of Ser to stimulate glycolysis under basal conditions and following citrullination. This is further supported by the observation that the ability of citrullination to overcome Phe inhibition through shifting the equilibrium towards Ser was also diminished by FBP. Together these observations support the idea that by disrupting its hydrogen bonding, R489 citrullination could act to lower FBP binding and its ability to inhibit Ser, while citrullination of R106 reduced inhibition by Phe/Ala/Trp shifting the equilibrium towards Ser. These two concerted citrullination events, may therefore reprogramme PKM2 to be principally regulated by Ser (Fig. 6e).

**Citrullination stimulates glycolysis in hypoxia**. The above analyses of the CCLE showed coordinate *PADI1* and *PADI3* expression in multiple cancer cell lines. Analyses of TCGA human tumour datasets showed strongly positively correlated co-expression of

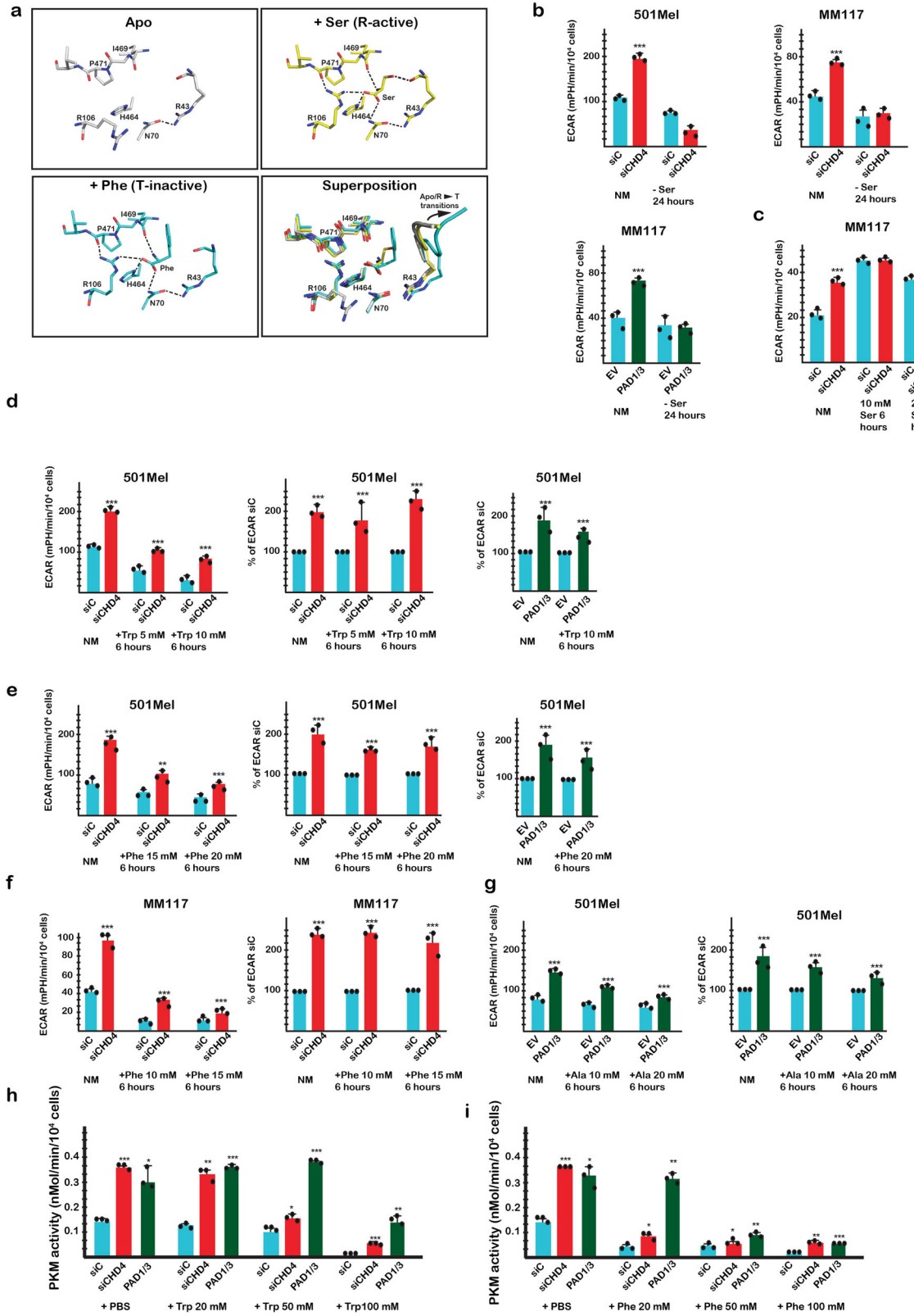

*PADI1* and *PADI3* in multiple tumour types such as cutaneous melanoma, uveal melanoma, bladder, lung and head and neck, with *PADI1* often being the gene showing strongest correlation with *PADI3* (Supplementary Fig. 10a–f). Despite this strong co-regulation, no negative correlation with *CHD4* was seen suggesting there may be alternative mechanisms regulating *PADI1* and *PADI3* co-expression in tumours.

PAD expression is up-regulated under hypoxic conditions, for example in glioblastoma[35]. In support of this, CCLE interrogation with DepMap revealed a positive correlation of *PADI1* and *PADI3* expression with several hypoxia signatures (Supplementary Fig. 10g, h). Interrogation of the TCGA showed *PADI1* and/or *PADI3* expression in several solid tumours, bladder, pancreatic, cervical, head and neck and clear cell renal cancers with

**Fig. 5 PKM2 citrullination diminishes allosteric inhibition by Phe/Ala/Trp. a** Close up view of free Ser and Phe interactions within the free amino acid binding pocket in the Apo, R-active and T-inactive states with a superposition of the three structures. All residues displayed are shown as sticks. In the superposition, the peptide bearing R43 is represented as ribbon to show the allosteric changes created upon Phe binding. Salt bridges and hydrogen bonds are shown as dashed lines. For clarity, the side chain of Phe 470, which stacks on R106 side chain, is not displayed. PDB data sets are as described in Supplementary Fig. 3. **b** ECAR values in absence of Ser after CHD4 silencing or PAD1/3 expression in 501Mel or MM117 cells; NM = normal medium. **c** ECAR values in presence of exogenous Ser with or without CHD4 silencing in 501Mel cells. **d, e** ECAR values in presence of exogenous Trp or Phe with or without CHD4 silencing or PAD1/3 expression in 501Mel cells. **f** ECAR values in presence of exogenous Phe with or without CHD4 silencing in MM117 cells. **g** ECAR values in presence of exogenous Ala with or without PAD1/3 expression in 501Mel cells. **h, i** PKM2 enzymatic activity in cell extracts supplemented with the indicated concentrations of Trp and Phe or PBS as control. In all experiments ECAR values and PKM2 enzymatic activity were determined from $n = 3$ biological replicates with 6 technical replicates for each N for ECAR. $n = 3$ biological replicates. and unpaired $t$-test with two tailed $p$-value analyses and confidence interval 95% were performed by Prism 5. $p$-Values: *$p < 0.05$; **$p < 0.01$; ***$p < 0.001$. Data are mean ± SEM. Values for PKM2 enzymatic activity were determined by Prism 5 using a 2-way ANOVA test.

known hypoxic character. We examined the correlation between the hypoxic signatures of several cancer types with *PADI1* and *PADI3* expression. In pancreatic adenocarcinoma, ccRCC, and lung adenocarcinoma, *PADI1* and *PADI3* expression was positively correlated with the hypoxic signature of the different patient samples (Fig. 7a–c). This correlation was lower in cancers, such as bladder orf cervical that had higher intrinsic *PADI1* and *PADI3* expression. These data indicated hypoxia rather than CHD4 as a major regulator of *PADI1* and *PADI3* expression in human tumours.

Glycolysis is increased under hypoxic conditions suggesting that increased PKM2 activity due to higher PAD1 and PAD3 expression may contribute to this effect. To test this idea, we grew 501Mel melanoma cells in hypoxic conditions where *PADI1* and *PADI3* expression was induced concomitantly with the hypoxia responsive gene VEGF (Fig. 7d). Similarly, *VEGF*, *PADI1* and *PADI3* expression was induced in pseudo-hypoxic conditions in presence of 300 μM dimethyloxalylglycine (DMOG) and when grown as 3D melanospheres where cells within the sphere are in hypoxia (Fig. 7e, f). As a consequence of induced PAD1/3 expression, citrullination of PKM2 R106 was increased in presence of DMOG, while that of R246 was unchanged (Fig. 7g, h). Glycolysis was also increased in presence of DMOG (Fig. 7i) and this increase was attenuated after siRNA knockdown of PAD1 and PAD3 as was the increased citrullination of R106. Hence the increased glycolysis seen in cells grown with DMOG where PAD1 and PAD3 expression was induced was at least in part mediated by PKM2 citrullination.

## Discussion

Here we describe a regulatory pathway by which PKM2 citrullination regulates glycolysis and cancer cell proliferation. PKM2 is an allostatic regulator integrating a finely balanced feedback mechanism that modulates its activity over a wide range of absolute and relative amino acid concentrations[9].

FBP and Ser each stimulate PKM2 activity by stabilising the active R-state[9]. Our data showed that exogenous FBP did not stimulate glycolysis in agreement with the report of Macpherson et al.[10], that intracellular FBP concentrations are sufficient to saturate PKM2. They also reported that FBP and Phe can simultaneously bind PKM2 with Phe preventing maximal activation of the FBP bound tetramer[10] maintaining PKM2 in a lower activity state as seen in tumours[4]. Glycolysis was stimulated by exogenous Ser. Stabilisation of the active state by Ser, whose binding is mutually exclusive with Phe/Trp/Ala, would therefore lead to higher PKM2 activity compared to FBP. Ser depletion lowered basal glycolysis consistent with a dynamic equilibrium between a Ser-bound PKM2 and a less active FBP-Phe form that limits glycolysis and allows its dynamic regulation by changing this equilibrium (Fig. 6e). In agreement with this, exogenous Ser stimulated glycolysis, pushing the equilibrium towards Ser bound

PKM2, whereas FBP that is already saturating and antagonised by Phe did not (Fig. 6e). Excess FBP did however counteract inhibition of glycolysis by exogenous Phe again consistent with their reported antagonism. Hence, while in vitro studies showed that FBP and Ser each stimulate PKM2 activity by stabilising the active R-state[9], our data show that FBP inhibited stimulation of glycolysis by Ser in vivo. This is unexpected given that FBP and Ser can bind PKM2 simultaneously at least under in vitro conditions used for crystallography. Our observations rather suggest that in vivo, the active R-state is stabilised by one or the other, but not by both simultaneously. How FBP antagonises stimulation by Ser remains to be determined.

The dynamic equilibrium regulating basal glycolysis can be upset by excess Ser but also by citrullination. Although we could not confirm R489 citrullination using specific antibodies, the results obtained using exogenous FBP and Ser were consistent with this modification. If R489 were to be citrullinated, the diminished hydrogen bonding with FBP would alleviate its negative effect on Ser-dependent stimulation of glycolysis. This effect is amplified by R106 citrullination that lowered PKM2 sensitivity to Trp/Ala/Phe shifting the equilibrium towards Ser. These two modifications could therefore act in concert to promote PKM2 regulation by Ser leading to increased glycolysis, an effect analogous to addition of excess Ser and in agreement with the observation that stimulation by citrullination required Ser (Fig. 6e).

While we detected citrullination of multiple enzymes of the glycolytic pathway, our data on the effects of Ser, Phe/Ala/Trp and FBP on glycolysis and/or PKM2 enzymatic activity all converge on PKM2 as being the central target. This was further supported by the observation that glycolysis in primary WI-38 fibroblasts that express PKM1 was not affected by ectopic PADI1/3 expression or addition of Phe and Trp and that PKM1 activity was not affected by siCHD4 silencing. However, we cannot exclude that citrullination of other glycolytic enzymes is also important. It has been reported that activity of ENO can be modulated by citrullination[30]. Nevertheless, the many publically available structures of PKM2 in different states allowed us to make and test predictions as to how citrullination of key arginines affects its activity. It will be interesting to assess how citrullination affects structure and function of the other glycolytic enzymes and to understand how this contributes to cellular metabolism. Similarly, although we cannot exclude that other genes deregulated by CHD4 silencing contribute to the observed cellular phenotypes, the de-regulated expression of PAD1 and/or PAD3 was a common feature in all the cell types tested and correlated with observed increase in glycolysis.

Our data provide insight as to how conversion of arginine to citrulline impacts their key interactions. Unlike other post-translational modifications such as phosphorylation, or methylation, and to some extent acetylation, that often act positively to

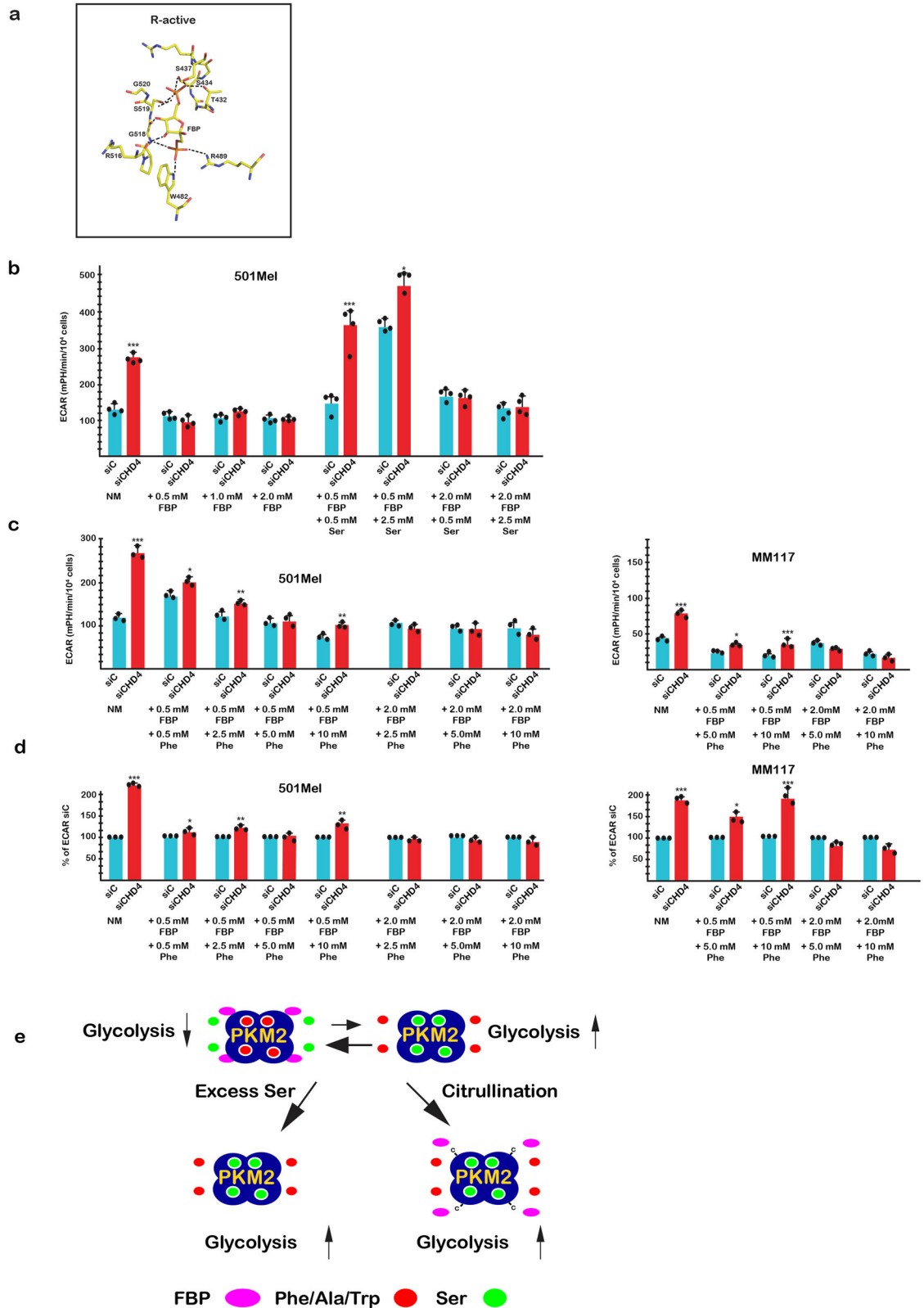

create new interactions with proteins that specifically recognise the modified amino acids, citrullination acts negatively due to loss of side chain charge and weakened hydrogen bonding ability. In the case of PKM2, our data illustrate how weakening of interactions paradoxically translated into a positive reprograming and stimulation of glycolysis.

PKM2 has been shown to be regulated by other post-translational modifications, the best characterised of which are tyrosine phosphorylation[12], lysine acetylation on K305[36] and K433[37] and oxidation of C358[38,39]. In each case, these modifications result in inhibition of PKM2 enzymatic activity. Moreover, most of the above studies concentrated on how post-translational

**Fig. 6 Effects of FBP and Ser on glycolysis. a** Close up view of FBP interactions with the R-active state illustrating the hydrogen bond between R489 and the 1′ phosphate of FBP as well as the network of hydrogen bonding with other residues. Salt bridges and hydrogen bonds are shown as dashed lines. For clarity, the side chain of K433 is not displayed. PDB data sets are as described in Supplementary Fig. S3 **b** ECAR values in presence of increasing exogenous FBP with or without exogenous Ser and siCHD4 silencing. NM = normal medium. **c** ECAR values in presence of increasing exogenous Phe with or low or high exogenous FBP and siCHD4 silencing in 501Mel or MM117 cells as indicated. **d** Effect of siCHD4 silencing on ECAR values in presence of increasing exogenous Phe with or low or high exogenous FBP expressed as a % of the siC control. $n = 3$ biological replicates with 6 technical replicates for each N. Unpaired $t$-test analysis were performed by Prism 5. $p$-Values: $*p < 0.05$; $**p < 0.01$; $***p < 0.001$. Data are mean ± SEM. **e** A model for how citrullination affects PKM2 and glycolysis. Under basal conditions PKM2, represented as a tetramer, is in a dynamic equilibrium between a Ser bound form and a lower activity FBP bound form also in equilibrium with inhibitory amino acids. Increased Ser shifts the equilibrium to a Ser-bound form with higher activity due to mutually exclusive occupancy by Ser or Trp/Phe/Ala accounting for the observed increase in glycolysis. Citrullination diminishes FBP binding (R489 < C represented by –C) alleviating its negative effect on Ser and shifts the mutually exclusive Ser vs Trp/Phe/Ala binding in favour of Ser. The net result is to promote a predominantly Ser-bound form accounting for the observed increased in glycolysis.

modifications affected PKM2 activity after cell lysis or PKM2 immunoprecipitation overlooking that PKM2 activity in cells is regulated by a dynamic and complex crosstalk amongst its different ligands[10]. Measuring glycolysis in the living cells was essential to assess how citrullination impacted cross-talk by multiple ligands to stimulate PKM2 and glycolysis. Citrullination is therefore a physiological mechanism that has an effect analogous to synthetic small molecules that increase PKM2 activity and stimulate excessive glycolysis resulting in Ser auxotrophy and reduced cell proliferation[5,6,40,41].

Under most normal conditions, expression of PAD1 and PAD3 is tightly regulated with low or no expression. In cancer cell lines, experimental CHD4 silencing and correlative expression over the CCLE both showed that CHD4 negatively regulated their expression. In several human tumours, *PADI1* and *PADI3* expression was positively correlated with hypoxia. Thus, hypoxia rather than CHD4 seems to be a major regulator of their expression in human tumours. We confirmed up-regulation of *PADI1* and *PADI3* in cells grown under hypoxic conditions and we further showed that the increased glycolysis under hypoxic conditions was attenuated by PAD1 and PAD3 silencing. Hence, PKM2 citrullination is enhanced under hypoxic conditions and contributes to the increased glycolysis.

PAD enzyme expression is also de-regulated in pathological situations such as rheumatoid arthritis (RA) where the production of antibodies against aberrantly citrullinated proteins contributes to the chronic inflammatory state[30,42,43]. Moreover, citrullination of glycolytic enzymes including PKM2 was observed in RA[30]. This study also showed that PAD1 and PAD3-mediated citrullination stimulated PKM2 enzymatic activity in vitro independent of FBP, Ser and inhibitory amino acids. We therefore cannot exclude that PKM2 citrullination has an intrinsic stimulatory activity in addition to modulating regulation by activator or inhibitor ligands. Moreover, the RA-associated environment is characterised by hypoxia and heterogeneous availability of nutrients, resembling that of some tumours[44]. Thus, PAD1 and PAD3 expression and the subsequent PKM2 citrullination seen in RA may account for the increased glycolysis seen in activated RA-associated fibroblast-like synoviocytes, another hallmark of the disease[44–46].

In conclusion, we identify a pathway regulating cancer cell proliferation where PAD1 and PAD3 citrullinate key arginines in PKM2 involved in its allosteric regulation to modulate glycolysis and cell proliferation. PAD1 and PAD3-mediated PKM2 citrullination also contributes to the increased glycolysis seen under hypoxic conditions, a hallmark of many cancers and RA and may be active in other pathological contexts associated with increased glycolysis.

## Methods
A list of oligonucleotides, antibodies and resources can be found in Supplementary Dataset 3.

**Cell culture, siRNA silencing and expression vector transfection**. Melanoma cell lines 501Mel and SK-Mel-28 were grown in RPMI 1640 medium supplemented with 10% foetal calf serum (FCS). MM074 and MM117 were grown in HAM-F10 medium supplemented with 10% FCS, 5.2 mM glutamax and 25 mM Hepes. Hermes-3A cell line was grown in RPMI 1640 medium (Sigma) supplemented with 10% FCS, 200 nM TPA, 200pM cholera toxin, 10 ng/ml human stem cell factor (Invitrogen) and 10 nM endothelin-1 (Bachem). HeLa cells were grown in Dulbecco's modified Eagle's medium supplemented with 10% FCS. SiHA cells were grown in EAGLE medium supplemented with 10% FCS, 0.1 mM non-essential amino acids and 1 mM sodium pyruvate. UOK cell lines were cultured in DMEM medium (4.5 g/L glucose) supplemented with 10% heat-inactivated FCS and 0.1 mM AANE. The WI-38 cell line was grown in MEM medium (Invitrogen) supplemented with 10% FCS, 0.1 mM AANE and 1 mM sodium pyruvate. For hypoxia experiments, cells were treated with DMOG (Sigma-Aldrich) for 24 h in normal culture conditions.

SiRNA knockdown experiments were performed with the corresponding ON-TARGET-plus SMARTpools purchased from Dharmacon Inc. (Chicago, IL, USA). SiRNAs were transfected using Lipofectamine RNAiMax (Invitrogen, La Jolla, CA, USA) and cells were harvested 72 h after. PADI1 and PADI3 expression vectors were transfected using X-tremeGENE 9 DNA Transfection Reagent (Sigma) for 48 h. To assess clonogenic capacity, cells were counted and seeded in 6 well plates for 7 to 15 days.

**Proliferation, viability and senescence analyses by flow cytometry**. To assess proliferation after siRNA treatment, cells were stained with Cell Trace Violet (Invitrogen) on the day of transfection. To assess cell viability, cells were harvested 72 h after siRNA transfection and stained with Annexin-V-FITC (Biolegend) following manufacturer instructions. Cells were analysed on a LSRII Fortessa (BD Biosciences) and data were analysed using Flowjo software v 6.8.

**ATP measurement**. The concentration of ATP was determined 72 h after siRNA transfection using the luminescent ATP detection system (Abcam, ab113849) following the manufacturer's instructions.

**Protein extraction and Western blotting**. Whole cell extracts were prepared by the standard freeze-thaw technique using LSDB 500 buffer (500 mM KCl, 25 mM Tris at pH 7.9, 10% glycerol (v/v), 0.05% NP-40 (v/v), 16 mM DTT, and protease inhibitor cocktail). Cell lysates were subjected to SDS–polyacrylamide gel electrophoresis (SDS-PAGE) and proteins were transferred onto a nitrocellulose membrane. Membranes were incubated with primary antibodies in 5% dry fat milk and 0.01% Tween-20 overnight at 4 °C. The membrane was then incubated with HRP-conjugated secondary antibody against rabbit or mouse as appropriate (Jackson ImmunoResearch) (1/5000 dilution in PBS) for 1 h at room temperature, and visualised using the ECL detection system (GE Healthcare). Antibodies: CHD3 abcam ab84528, CHD4 abcam ab72418, MITF abcam ab3201, SOX10, Abcam, ab155279.

**Pyruvate kinase enzyme activity**. Cells were seeded in 6-well plates and transfected with siRNAs for 72 h or expression vectors for 24 h. The cells were lysed with RIPA buffer, and pyruvate kinase activity in the cell lysates was measured by the PK assay kit (Sigma, MAK072) according to the manufacturer's instructions. All the values were normalised to cell numbers.

**Generation of anti-Cit106 and anti-Cit246 PKM2 antibodies**. Antibodies against citrulline-containing peptides were raised in rabbits by the BioGenes company using the peptide sequences SFASDPILY-CIT-PVAVALDTKGGC and ASFI-Cit-KASDVHEVRKVLGEGGC for R106Cit and R246Cit respectively. Peptides were generated, quantified and confirmed by mass spectrometry by Genscript. Rabbits were immunised with carrier-conjugated peptide followed by three 3 booster injections after 14, 28 and 42 days. Affinity purification of antisera were performed by coupling the citrullinated peptides or their wild-type counterparts to SulfoLink Coupling Gel

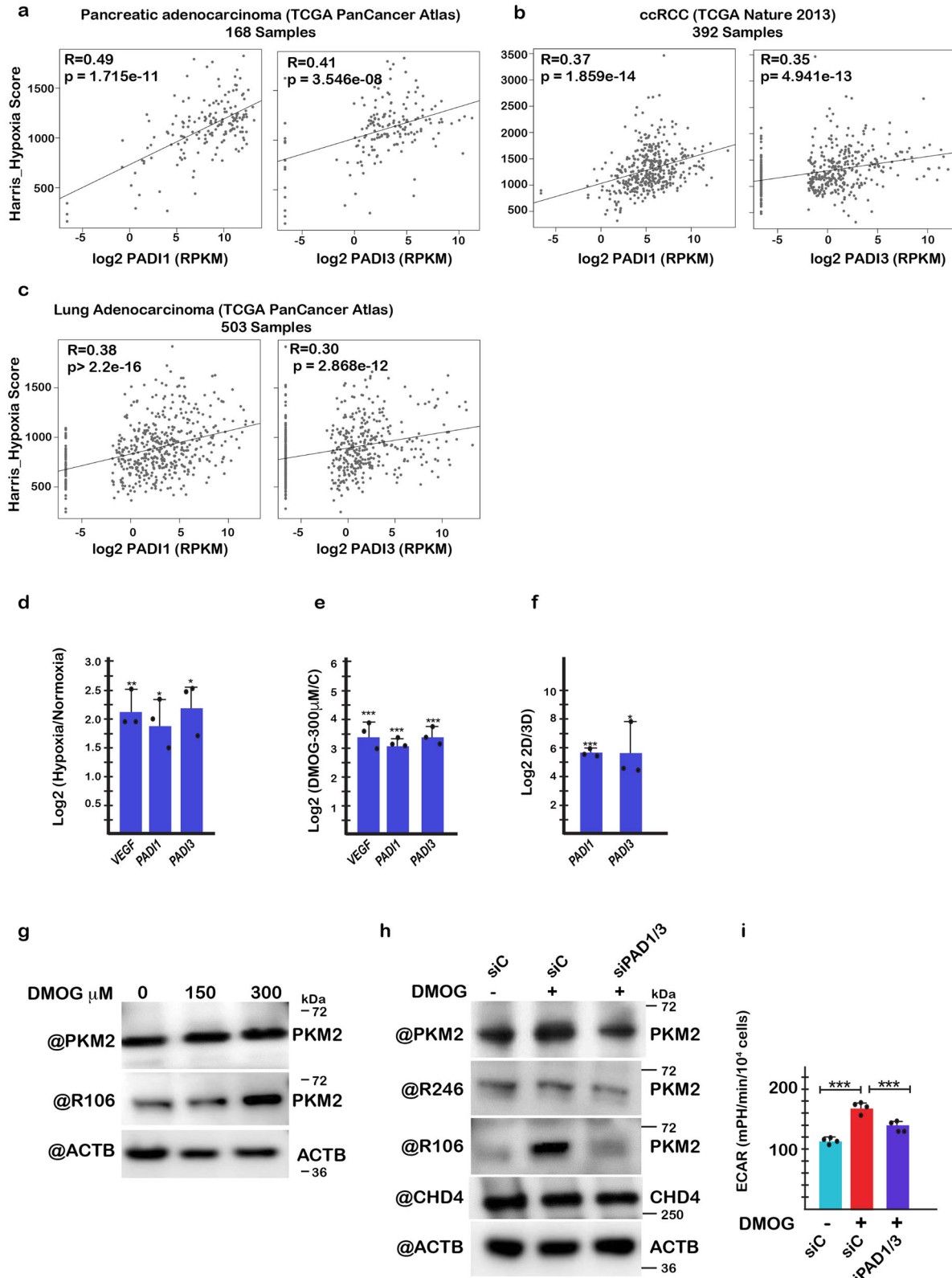

**Fig. 7 PKM2 citrullination contributes to increased glycolysis in hypoxia. a–c** Correlation of *PADI1* and *PADI3* expression with that of the HARRIS_HYPOXIA signature in the indicated cancer types using TCGA data sets. The Spearman correlations and *p*-values are indicated. **d, e** Expression of the indicated genes assessed by RT-qPCR in cells grown under hypoxia (1% O$_2$) (**d**) or pseudo-hypoxia with 300 μM DMOG (**e**). *n* = 3 biological replicates. **f** Induction of PAD1 and PAD3 expression in 501Mel cells grown as 3D melanospheres for 7 days. **g** Immunoblots showing enhanced PKM2 R106Cit when grown under increasing concentrations of DMOG. **h** Immunoblots showing expression of the indicated proteins in transfected cells in presence of 300 μM DMOG. **i** Glycolysis in transfected cells in presence of 300 μM DMOG. ECAR values were determined from *n* = 4 biological replicates with 6 technical replicates for each N and unpaired *t*-test with two tailed *p*-value analyses and confidence interval 95% were performed by Prism 5. *p*-Values: *\*p* < 0.05; \*\**p* < 0.01; \*\*\**p* < 0.001. Data are mean ± SEM.

(PIERCE, 20401) agarose beads. The antisera were passed first through the column with citrullinated peptides and then through column with the wild-type peptide to remove residual antibodies recognising the wild-type peptide.

**Immunoprecipitation and mass-spectrometry.** Citrullinated proteins were immunoprecipitated from whole cell extracts with an anti-pan-citrulline antibody (Abcam, ab6464). Samples were concentrated on Amicon Ultra 0.5 mL columns (cutoff: 10 kDa, Millipore), resolved by SDS-PAGE and stained using the Silver 7 Quest kit (Invitrogen).

**Mass spectrometry and analysis.** Mass-spectrometry was performed at the IGBMC proteomics platform (Strasbourg, France). Samples were reduced, alkylated and digested with LysC and trypsin at 37 °C overnight. Peptides were then analysed with an nanoLC- MS/MS system (Ultimate nano-LC and LTQ Velos ion trap, Thermo Scientific, San Jose Califronia). Briefly, peptides were separated on a C18 nano-column with a 1 to 30% linear gradient of acetonitrile and analysed in a TOP20 CID data-dependent MS method. Peptides were identified with SequestHT algorithm in Proteome Discoverer 2.4 (Thermo Fisher Scientific) using Human Swissprot database (20347 sequences). Precursor and fragment mass tolerance were set at 0.9 Da and 0.6 Da respectively. Trypsin was set as enzyme, and up to 2 missed cleavages were allowed. Oxidation (M) and Citrullination (R) were set as variable modifications, and Carbamidomethylation (C) as fixed modification. Peptides were filtered with a 1 % FDR (false discovery rate) on peptides and proteins. For statistical analyses data was analysed using Perseus identifying PSMs and proteins that were enriched in the siCHD4 extracts[47].

**Chromatin immunoprecipitation and sequencing.** CHD4 ChIP experiments were performed on 0.4% Paraformaldehyde fixed and sonicated chromatin isolated from 501Mel cells according to standard protocols[48]. MicroPlex Library Preparation kit v2 was used for ChIP-seq library preparation. The libraries were sequenced on Illumina Hiseq 4000 sequencer as Single-Read 50 base reads following Illumina's instructions. Sequenced reads were mapped to the Homo sapiens genome assembly hg19 using Bowtie with the following arguments: -m 1 --strata --best -y -S -l 40 -p 2. After sequencing, peak detection was performed using the MACS software[49]. Peaks were annotated with Homer (http://homer.salk.edu/homer/ngs/annotation.html) using the GTF from ENSEMBL v75. Peak intersections were computed using bedtools and Global Clustering was done using seqMINER. De novo motif discovery was performed using the MEME suite (meme-suite.org). Motif enrichment analyses were performed using in house algorithms as described in[50].

**RNA preparation, quantitative PCR and RNA-seq analysis.** RNA isolation was performed according to standard procedure (Qiagen kit). qRT-PCR was carried out with SYBR Green I (Qiagen) and Multiscribe Reverse Transcriptase (Invitrogen) and monitored using a LightCycler 480 (Roche). RPLP0 gene expression was used to normalise the results. Primer sequences for each cDNA were designed using Primer3. RNA-seq was performed essentially as described[51]. Gene ontology analyses were performed with the Gene Set Enrichment Analysis software GSEA v3.0 using the hallmark gene sets of the Molecular Signatures Database v6.2 and the functional annotation clustering function of DAVID.

**Analysis of oxygen consumption rate (OCR) and glycolytic rate (ECAR) in living cells.** The ECAR and OCR were measured in an XF96 extracellular analyzer (Seahorse Bioscience). A total of 20000 cells per well were seeded and transfected by siRNA or expression vector 72 h and 24 h respectively prior to the experiment. The cells were incubated in a CO2-free incubator at 37 °C and the medium was changed to XF base medium supplemented with 1 mM pyruvate, 2 mM glutamine and 10 mM glucose for an hour before measurement. For OCR profiling, cells were sequentially exposed to 2 μM oligomycin, 2 μM carbonyl cyanide-4-(trifluoromethoxy) phenylhydrazone (FCCP), and 0.5 μM rotenone and antimycin A. For ECAR profiling, cells were sequentially exposed to 2 μM oligomycin and 150 mM 2-deoxyglucose (2-DG). After measurement, cells were washed with PBS, fixed with 3% PFA, permeabilized with 0.2% triton. Nuclei were counterstained with Dapi (1:500) and number of cells per well determined by the IGBMC High Throughput Cell-based Screening Facility (HTSF, Strasbourg). L-Phe (Sigma, P2126), L-Trp (Sigma, T0254), L-Ala (Sigma, A7627), L-Ser (Sigma, S4500) or D-FBP (Sigma, F6803) were added in the complete medium 24 h for Serine and 6 h for Trp/Phe/Ala/FBP and in the refreshed XF base medium prior the experiment.

**Statistics and reproducibility.** Values for all RT-qPCR analyses, cell based assays (proliferation, apoptosis, colony forming capacity and ATP levels) were performed from a minimum of $n = 3$ biological replicates. ECAR and OCR values from Seahorse experiments were derived from a minimum of $n = 3$ biological replicates with 6 technical replicates for each N. Unpaired $t$-test with two tailed $p$-value analyses and confidence interval of 95% were performed by Prism 5. $p$-Values: *$p < 0.05$; **$p < 0.01$; ***$p < 0.001$. PKM2 enzymatic assays were performed in duplicate or triplicate and analysed by Prism 5 using a 2-way ANOVA test. All immuno-blots had at least one independent repeat with similar results.

**Data mining.** For Figures S3a–c and S10a–f, we interrogated the Cancer Cell Line Encyclopedia or the indicated tumour data sets on the cBioPortal interface (http://www.cbioportal.org/) and used the co-expression function to calculate to correlations between the genes. For Figure S10g–h, we used the PanCancer Atlas selection for *PADI1* or *PADI3* as indicated. For Fig. 7a–c, we retrieved the normalised RNA-seq data for the different tumour types via the cBioPortal interface (http://www.cbioportal.org/) as follows: the data sets for the indicated tumours were first queried by gene for *PADI1* or *PADI3*. We followed the download command on the results page and downloaded the transposed matrix version of the mRNA expression (RNA Seq V2 RSEM) file. Expression of *PADI1* and *PADI3* was converted to log2(RPKM). To remove zeros, the value "0.01" was added to all RPKM values. Hypoxia signature score was defined as the geometric mean of the expression from the list of genes of the Msigdb Geneset HARRIS_HYPOXIA (https://www.gsea-msigdb.org/gsea/msigdb/cards/HARRIS_HYPOXIA).

Correlations between hypoxia signature and *PADI1*/*PADI3*expression were performed in R (v3.6.3) with the cor.test function using the Spearman method and represented as a scatterplot with the plot function.

For Figure S10i–j, correlations of *PADI1* and *PADI3* expression with hypoxic signatures over 1848 cancer cell lines was performed using DepMap (depmap.org/portal/). CCLE expression data was quantified online from RNA-seq files using the GTEx pipelines. A detailed description of the pipelines and tool versions can be found at: https://github.com/broadinstitute/ccle_processing#rnaseq. DepMap Public 20Q4.

**Reporting summary.** Further information on experimental design is available in the Nature Research Reporting Summary linked to this paper.

## Data availability

Source data are provided within this paper and are available from the authors upon reasonable request. The CHD4 ChIP-seq and RNA-seq data described here have been deposited in GEO with the accession number GSE134850 The additional ChIP-seq data used for Figure S2 are available in the GEO data database under accession codes: https://www.ncbi.nlm.nih.gov/geo/query/acc.cgi?acc=GSE94488, https://www.ncbi.nlm.nih.gov/geo/query/acc.cgi?acc=GSE94488 and https://www.ncbi.nlm.nih.gov/geo/query/acc.cgi?acc=GSM2842802. Source data are provided with this paper.

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

## Acknowledgements
We thank, Dr Goncalo Castelo-Branco for the PAD3 expression vector, J-C. Marine and G. Ghanem for the MM117 and MM074 primary melanoma cells, D. Bennet for the HERMES-3A line, all the staff of the IGBMC common facilities in particular the IGBMC mass spectrometry platform and the high throughput screening facility. This work was supported by institutional grants from the Centre National de la Recherche Scientifique, the Institut National de la Santé et de la Recherche Médicale, the Université de Strasbourg, the Association pour la Recherche contre le Cancer (CR, contract number PJA 20181208268), the Ligue Nationale contre le Cancer, the Institut National du Cancer, the ANR-10-LABX-0030-INRT French state fund through the Agence Nationale de la Recherche under the frame programme Investissements d'Avenir labelled ANR-10-IDEX-0002-02. The IGBMC high throughput sequencing facility is a member of the "France Génomique" consortium (ANR10-INBS-09-08). The mass spectrometry facility is supported by grants from the ARC foundation and from the Canceropole Grand Est. ID is an 'équipe labellisée' of the Ligue Nationale contre le Cancer. SC was supported by a fellowship from the Ligue Nationale contre le Cancer.

## Author contributions
S.C. performed ChIP-seq, RNA-seq, transfections and metabolism experiments, G.D. and G.G. performed bioinformatics analyses, L.N. performed and analysed mass-spectrometry experiments, S.D. constructed and provided PADI1 expression vector, C.R. performed structural analyses. S.C., S.D., C.R. and I.D. conceived the experiments, analysed the data and wrote the paper.

## Competing interests
The authors declare no competing interests.
