## [Peer Review File · Nature Communications]

Reviewers' comments:

Reviewer #1 (Remarks to the Author); expert on PKM2 and biochemistry:

This paper introduces the interesting idea that citrullination of pyruvate kinase M2 (up)regulates its enzyme activity and also modulates the effect of other PKM2 activators (Ser,FBP) and inhibitors (Phe,Ala, Trp).

The clear experimental result presented is that CHD4 (an ATPase domain of NuRD) acts as a repressor regulating the expression of PADI 1 and PADI 3. This seems to be well substantiated by RNA-seq experiments, though they are only 2 genes out of ~1000 that have been upregulated. The increase in citrullination is also clearly demonstrated by the near doubling of Peptide Spectral Matches for siCHD4 cells compared with control cells. Interestingly almost all the glycolytic enzymes also show significant enrichment in citrullination. It would be useful to mark them on the volcano plot and also identify some of the other highly enriched/significant proteins on the plot. The main problem for me is how to justify that out of these hundreds of citrullination events identified in the volcano plot, that the three PKM2 Arg residues are (solely) responsible for the changes in glycolysis which are clearly shown in the siCHD4 cells.

It is indeed intriguing that that two of the three identified citrullinated Arg residues of PKM2 are involved in direct binding to the effector molecules (R106 forms a salt bridge with the carboxyl group of the amino acid effectors) and R489 forms a salt bridge with FBP). Less convincing to my mind is the proposed effect of citrullination on the interaction between R246 and the two carbonyl oxygen atoms of V215 and L217. This is not really a salt bridge but rather two hydrogen bonds which could equally well be made with a citrulline residue.

The results on monitoring glycolysis in siCHD4 cells in the presence of the amino acids and FBP are also interesting and consistent with most of the proposed allosteric regulatory mechanisms identified in PKM2. However, given that there are only cellular readouts it is probably over interpreting the data to draw detailed conclusions regarding enzyme mechanism without some enzymatic data – ideally a citrullinated form of PKM2 or some site point mutant analogues. One possibly important point that needs a bit of discussion is that there is no real evidence provided to show that only the PKM2 isoform is relevant in these cellular studies. Is there evidence to show that there is no PKM1 in these cancer cell lines?

In summary, the paper presents some very interesting results and shows that citrullination is playing a major regulatory role in glycolysis. However given that the results are wholly dependent on cellular data, care should be taken not to over interpret the data and suggest that the changes in glycolysis are all down to PKM2 without providing some supporting enzymatic data on modified/mutant PKM2.

Reviewer #2 (Remarks to the Author); expert on citrullination:

Coassolo et al (2020) Citrullination of pyruvate kinase by PADI1 and PADI3 regulates glycolysis and cancer cell proliferation

In this very interesting manuscript Coassolo et al demonstrate that knockdown of CHD4, the ATPase subunit of the NuRD chromatin remodeling complex, leads to highly increased expression of PADI1 and PADI3 in a variety of cancer cell lines representative of melanoma as well as numerous other cancer types (e.g. breast, cervical). Proteomic methods identified several citrullinated proteins whose levels were increased upon induction of PADI1 and PADI3 expression. Amongst these proteins was PKM2 and the authors confirm that PKM2 is citrullinated at higher levels when CHD4 is knocked down and this is associated with increased glycolysis. Overall, the results are highly interesting and will be broadly appreciated, however, there are several issues that need to be addressed before this reviewer can recommend publication. These issues are described below:

1. The authors identified several citrullination sites (R106, R246, and R489) in PKM2 but do not show the MS and MS/MS data that confirms that they are indeed citrullinated. This data needs to be included. If they do not have site of modification data, they need to explicitly state that this is the case.

2. The methods describing the quantitative proteomics do not provide information on how they were performed. This information must be included.
3. The authors suggest that an arginine can form a salt bridge with backbone carbonyls. This type of ionic interaction is not possible; carbonyls are not charged. They can however form hydrogen bonds.
4. The authors spend much time discussing the roles of citrullinating specific arginines in PKM2 (e.g., R106 and R489) but the links between the citrullination of these residues and changes in glycolysis are only correlative as they are derived from experiments where CHD4 was knocked down or PADs 1 and 3 were overexpressed. As such, the authors should include in vitro assay data on mutants to validate that citrullination of these residues has the purported effects on binding of Serine or inhibitory amino acids.
5. The authors should note that Tilvawala et al (ref 41) showed that citrullination of PKM2 by PADs 1 and 3 led to an ~2-fold increase in in vitro PKM2 activity and this effect was independent of FBP binding.
6. The authors refer to the proteins as PADI1 and PADI3. Although this annotation is used by uniprot, the proper convention is to refer to the enzymes as PADs and the genes as PADIs. This nomenclature is widely used in the citrullination field, especially the rheumatology community, where much of the citrullination literature has been centered. The authors should therefore revise their manuscript so that it follows the generally accepted format.

Reviewer #3 (Remarks to the Author); expert on epigenetics and metabolism:

In the Coassolo et al study, the authors hypothesize that CHD4 regulates melanoma cell proliferation and metabolism via transcriptional regulation of PADI1/3 and subsequent citrullination of its substrates, such as PKM2. The study presents interesting phenotypes mediated by CHD4 and PADI1/3, which may be important to cancer cell metabolism. Indeed, CHD4 mutations have been observed in several types of cancer.

The overall role of CHD4 chromatin remodeling in cancer development is significant, and the authors do demonstrate that metabolism is altered in siCHD4 cells. However, the study falls short of presenting a cohesive mechanistic model. Many genes become deregulated upon loss of CHD4, it is not clear why the authors focused on PADI1/3. Moreover, it is not entirely clear why the authors focus on PKM2 as the central substrate of PADI1/3. Importantly, the study lacks convincing data that links PADI1/3 and citrullinated PKM2 in the phenotypes mediated by CHD4 loss. Substantial additional experiments, detailed below, would be needed to examine the model proposed by the authors.

Major points:

- More background on CHD4 mutation frequency in cancers, particularly melanoma, would add to the significance of the study. In addition, the prevalence of glycolytic metabolism in cancers with CHD4 mutation would also bolster significance of results in this manuscript.
- The focus on both CHD3 and CHD4 in the introduction and Fig. 1 is not clear as only CHD4 phenotypes are subsequently investigated. The manuscript may have a better logical flow if CHD3 results were deemphasized.
- The logic of focusing on PADI1 and 3 as critical mediators of CHD4 function is not clear. This chromatin-remodeler regulates a multitude of genes that can influence proliferation and the other phenotypes measured in this study. In order to better link PADI1/3 in these phenotypes, expression would need to be restored in siCHD4 cells.
- The mechanism of CHD4-regulated PADI1/3 expression is not entirely clear, as ChIP and RNA-seq data is limited to correlation, not causation. The authors should temper their conclusions that CHD4 prevents activation by CTCF or other transcription factors. More experiments in siCHD4 cells would be needed to make this conclusion.
- On a related note, the correlation with PADI1/3, CTCF, and CHD4 expression is interesting, but would be more informative if CHD4 mutation was also factored into these analyses. So, for example, in cancer cells with CHD4 mutation, is PADI1/3 expression altered?
- The citrullination data in fig 2 is interesting. However, again, the focus on PKM2 is not entirely clear. Can the authors present a broader view of what pathways are enriched/depleted in siCHD4

cells? Are known substrates of PADI1/3 recovered in this assay? Moreover, a rescue experiment with restoration of PADI1/3 expression is needed to link PADI1/3 function to CHD4-mediated citrullination.

- In the antibody characterization in Supp fig 4, more controls are needed, such as siPADI1/3 and siPKM2.

- For the metabolic assays in fig 3, more support is needed for the role of PADI1/3 and PKM2 citrullination in the phenotypes mediated by CHD3. More assays should be performed rescuing the siCHD4 phenotype and with PKM2 citrullination mutants.

- It would seem that the alteration of media supplements in fig 5 would have a broad transcriptional and metabolic effect on the cell. Can the authors elaborate on why the observed changes are influenced by PKM2 citrullination, rather than other factors?

Minor points

- What is the designation of "slow proliferating cells" in Fig. 1? How is slow vs fast measured?

- For supplemental fig 3, there appear to be many possible outlier data points. Are these excluded when calculated correlation values?

- Can the authors comment on why fig 3g and k are not proportional to the level of PKM2 citrullination in fig 2?

Reviewers' comments:

We thank the referees for their constructive comments. We have performed a number of new experiments in response to their comments. A detailed point-by-point response is given below.

Reviewer #1 (Remarks to the Author); expert on PKM2 and biochemistry:

'Interestingly almost all the glycolytic enzymes also show significant enrichment in citrullination. It would be useful to mark them on the volcano plot and also identify some of the other highly enriched/significant proteins on the plot.'

We have made a new version of Figs. 2e and f to indicate the glycolytic enzymes on the volcano plot and their enrichment in the siCHD4 samples. For the sake of clarity, we did not indicate any other proteins as panel E is already rather crowded. We reanalyzed the data with the latest updated version of SequestHT algorithm in Proteome Discoverer 2.4 and updated Supplemental dataset 2 with the corresponding Perseus output. The new analyses identified some additional proteins not seen in the original analyses and showed an increased number of citrullinated PSMs in the siCHD4 sample thereby strengthening our conclusions. The data for PKM2 were unchanged. We added a new supplemental figure (Fig. S4) that shows the KEGG pathway analyses of the proteins that are enriched in the siCHD4 samples and the overlap between citrullinated proteins found in this study as enriched in the siCHD4 samples and those previously found in other studies from human and mouse tissues as well as rheumatoid arthritis. These previous studies are cited in the revised manuscript.

'The main problem for me is how to justify that out of these hundreds of citrullination events identified in the volcano plot, that the three PKM2 Arg residues are (solely) responsible for the changes in glycolysis which are clearly shown in the siCHD4 cells.'

The referee raises an important point. We performed a number of additional experiments to address this issue. 1). We analyzed the effect of siCHD4 and ectopic expression of PADI1, PADI3 or both on PKM2 enzymatic activity. This new data shows stimulation of PKM2 enzymatic activity in each condition. Furthermore, we also analyzed PKM2 enzymatic activity in the presence of increasing quantities of Phe and Trp. We found that siCHD4 and ectopic PADI expression partially overcame the inhibitory effect of Phe and Trp in a manner analogous to what was seen with glycolysis. This new data shows that citrullination stimulates PKM2 enzymatic activity and that it diminishes the sensitivity of

the enzyme to inhibitory amino acids. Thus, the changes in glycolysis are mirrored by the changes in activity of the PKM2 enzyme consistent with the extensive current literature on the key role of PKM2 in regulation of glycolysis and its regulation by the inhibitory amino acids. This new data is shown in Figs. 3f, 5h-i and in a new supplemental Fig. 6. 2). We showed that primary WI-38 fibroblasts as expected express PKM1 and not PKM2. We showed that unlike cancer cells glycolysis in WI-38 cells is insensitive to PADI1/3 expression and exogenous Phe and Trp and that PKM1 activity is insensitive to siCHD4 silencing. Hence, sensitivity of cells to siCHD4 and presence of inhibitory amino acids correlates with the presence of PKM2. This new data is shown in supplemental Figs. 5h and 6.

Together these additional experiments strongly point towards PKM2 as the key enzyme principally responsible for the altered glycolysis. However, as mentioned in the original version of the manuscript, we cannot exclude that citrullination of other glycolytic enzymes is also important. Indeed, there is evidence that the activities of NNMT and ENO are modulated by citrullination (cited in revised text). We are fortunate in that there are many publically available structures of PKM2 in different states allowing us to make and test predictions as to how citrullination of key arginines affects its activity. Much more work will be required to assess how citrullination may affect structure and function of the other glycolytic enzymes and how the changes affect cellular metabolism. We have stressed these issues in the revised version of the discussion.

'It is indeed intriguing that that two of the three identified citrullinated Arg residues of PKM2 are involved in direct binding to the effector molecules (R106 forms a salt bridge with the carboxyl group of the amino acid effectors) and R489 forms a salt bridge with FBP). Less convincing to my mind is the proposed effect of citrullination on the interaction between R246 and the two carbonyl oxygen atoms of V215 and L217. This is not really a salt bridge but rather two hydrogen bonds which could equally well be made with a citrulline residue.'

We agree with the referee and we have changed the text to take this into account.

The results on monitoring glycolysis in siCHD4 cells in the presence of the amino acids and FBP are also interesting and consistent with most of the proposed allosteric regulatory mechanisms identified in PKM2. However, given that there are only cellular readouts it is probably over interpreting the data to draw detailed conclusions regarding enzyme mechanism without some enzymatic data – ideally a citrullinated form of PKM2 or some site point mutant analogues.

We agree with the referee, and as described above we analyzed PKM2 enzymatic activity in addition to glycolysis.

‘One possibly important point that needs a bit of discussion is that there is no real evidence provided to show that only the PKM2 isoform is relevant in these cellular studies. Is there evidence to show that there is no PKM1 in these cancer cell lines?’

Again, we agree with the referee and as described above, the new data show that low levels of PKM1 are found in some of the cancer cell lines, but that they express higher levels of PKM2. This is consistent with the current literature. For example, we analyzed PKM-isoform usage in the 501Mel RNA-seq data. In the table shown below, the PKM1 isoform is much less abundant than the PKM2 isoform.

Transcript_id	Gene_id	Gene Name	Isoform	Length	Effective_length	Expected_count	TPM	FPKM	IsoPct
ENST00000335181	ENSG00000067225	PKM	PKM2	2318	2146,69	60335,42	1187,46	867,09	86,31
ENST00000449901	ENSG00000067225	PKM	PKM2	2254	2082,69	17,39	0,35	0,26	0,03
ENST00000567118	ENSG00000067225	PKM	PKM2	2254	2082,69	17,39	0,35	0,26	0,03
ENST00000568857	ENSG00000067225	PKM	PKM2	1355	1183,69	1164,54	41,57	30,35	3,02
								Total:	89,39
ENST00000389093	ENSG00000067225	PKM	PKM1	2302	2130,69	1194,76	23,69	17,3	1,72
ENST00000561609	ENSG00000067225	PKM	PKM1	1568	1396,69	0	0	0	0
ENST00000565154	ENSG00000067225	PKM	PKM1	2004	1832,69	0	0	0	0
ENST00000565184	ENSG00000067225	PKM	PKM1	2503	2331,69	321,21	5,82	4,25	0,42
ENST00000568459	ENSG00000067225	PKM	PKM1	1806	1634,69	0	0	0	0
ENST00000568883	ENSG00000067225	PKM	PKM1	1864	1692,69	0	0	0	0
								Total:	2,14

In contrast, primary WI-38 fibroblasts do not express PKM2 and are not sensitive to siCHD4 silencing and inhibitory amino acids. This is consistent with a wealth of literature cited in our paper that shows that PKM1 is not regulated by activating or inhibiting ligands.

‘In summary, the paper presents some very interesting results and shows that citrullination is playing a major regulatory role in glycolysis. However, given that the results are wholly dependent on cellular data, care should be taken not to over interpret the data and suggest that the changes in glycolysis are all down to PKM2 without providing some supporting enzymatic data on modified/mutant PKM2’.

In summary, we have addressed the major issues by addition of enzymatic data and controls with cells expressing PKM1.

Reviewer #2 (Remarks to the Author); expert on citrullination:

Coassolo et al (2020) Citrullination of pyruvate kinase by PADI1 and PADI3 regulates glycolysis and cancer cell proliferation

Overall, the results are highly interesting and will be broadly appreciated, however, there are several issues that need to be addressed before this reviewer can recommend publication. These issues are described below:

1. The authors identified several citrullination sites (R106, R246, and R489) in PKM2 but do not show the MS and MS/MS data that confirms that they are indeed citrullinated. This data needs to be included. If they do not have site of modification data, they need to explicitly state that this is the case.

For the referee, we include here the data from the MS/MS analysis that identifies the modified arginines. This data has been included in the 'Source data file'.

TATESFASDPILYRPVAVALDTK, R14-Deamidated (0.98402 Da)
 Charge: +3, Monoisotopic m/z: 822.53379 Da (-229.93 mmu/-279.54 ppm), MH+: 2465.58681 Da, RT: 76.8290 min, Identified with: Sequest HT (v1.17); XCorr:2.00, Percolator PEP:5.8e-2, ptmRS: Best Site Probabilities:R14(Deamidated): 100

#1	b	b ²	Seq.	y	y ²	#2
1	102.1	51.5	T			23
2	173.1	87.0	A	2365.2	1183.1	22
3	274.1	137.6	T	2294.2	1147.6	21
4	403.2	202.1	E	2193.1	1097.1	20
5	490.2	245.6	S	2064.1	1032.6	19
6	637.3	319.1	F	1977.1	989.0	18
7	708.3	354.7	A	1830.0	915.5	17
8	795.4	398.2	S	1759.0	880.0	16
9	910.4	455.7	D	1671.9	836.5	15
10	1007.4	504.2	P	1556.9	779.0	14
11	1120.5	560.8	I	1459.9	730.4	13
12	1233.6	617.3	L	1346.8	673.9	12
13	1396.7	698.8	Y	1233.7	617.3	11
14	1553.7	777.4	R-deam	1070.6	535.8	10
15	1650.8	825.9	P	913.5	457.3	9
16	1749.9	875.4	V	816.5	408.7	8
17	1820.9	911.0	A	717.4	359.2	7
18	1920.0	960.5	V	646.4	323.7	6
19	1991.0	996.0	A	547.3	274.2	5
20	2104.1	1052.6	L	476.3	238.6	4
21	2219.1	1110.1	D	363.2	182.1	3
22	2320.2	1160.6	T	248.2	124.6	2
23			K	147.1	74.1	1

Sequence: FGVEQDVMVFASFIR, R16-Deamidated (0.98402 Da)
 Charge: +2, Monoisotopic m/z: 931.21995 Da (+274.71 mmu/+295 ppm), MH+: 1861.43262 Da, RT: 106.5405 min, Identified with: Sequest HT (v1.17); XCorr:4.54, Percolator PEP:2.4e-5, ptmRS: Best Site Probabilities:R16(Deamidated): 100

#1	b	Seq.	y	#2
1	148.1	F		16
2	205.1	G	1712.8	15
3	304.2	V	1655.8	14
4	433.2	E	1556.7	13
5	561.3	Q	1427.7	12
6	676.3	D	1299.6	11
7	775.4	V	1184.6	10
8	890.4	D	1085.5	9
9	1021.4	M	970.5	8
10	1120.5	V	839.5	7
11	1267.6	F	740.4	6
12	1338.6	A	593.3	5
13	1425.6	S	522.3	4
14	1572.7	F	435.3	3
15	1685.8	I	288.2	2
16		R-deam.	175.1	1

Sequence: DPVQEAWAEDVDLR, R14-Deamidated (0.98402 Da)
 Charge: +2, Monoisotopic m/z: 822.12729 Da (-253.8 mmu/-308.71 ppm), MH+: 1643.24731 Da, RT: 75.2005 min, Identified with: Sequest HT (v1.17); XCorr:4.46, Percolator PEP:4.0e-4, ptmRS: Best Site Probabilities:R14(Deamidated): 100

#1	b	Seq.	y	#2
1.0	116.0	D		14.0
2.0	213.1	P	1528.7	13.0
3.0	312.2	V	1431.7	12.0
4.0	440.2	Q	1332.6	11.0
5.0	569.3	E	1204.5	10.0
6.0	640.3	A	1075.5	9.0
7.0	826.4	W	1004.5	8.0
8.0	897.4	A	818.4	7.0
9.0	1026.5	E	747.4	6.0
10.0	1141.5	D	618.3	5.0
11.0	1240.5	V	503.3	4.0
12.0	1355.6	D	404.2	3.0
13.0	1468.7	L	289.2	2.0
14.0		R-Deam.	176.1	1.0

We note and cite in the revised version of the paper that 2 of the 3 sites we identified (R016 and R246) were previously found in other studies of citrullinated proteins in human or mouse tissues or in rheumatoid arthritis. We confirmed the MS data with site specific anti-R106Cit and R246Cit antibodies. Thus, in our data, R106 and R246 the best site probabilities correspond to residues already identified in other studies and corroborated here using

specific antibodies. Unfortunately, we were not able to generate peptides and antibodies corresponding to R489Cit, but the best site probability for this residue is comparable to that of the others giving us confidence that this residue is also a *bone fide* target of citrullination.

'2. The methods describing the quantitative proteomics do not provide information on how they were performed. This information must be included.'

As we mentioned in the Methods section, the statistical analyses of the data (rather than quantitative proteomics, a term we did not use) was performed using the Perseus package. We do not claim to have performed quantitative proteomics per se (such as Silac for example), but rather to a statistical analysis of the data using Perseus. The output file is included as Supplemental dataset 2.

3. The authors suggest that an arginine can form a salt bridge with backbone carbonyls. This type of ionic interaction is not possible; carbonyls are not charged. They can however form hydrogen bonds.

We agree with the referee (see also above, referee 1) and we have modified the text accordingly.

'4. The authors spend much time discussing the roles of citrullinating specific arginines in PKM2 (e.g., R106 and R489) but the links between the citrullination of these residues and changes in glycolysis are only correlative as they are derived from experiments where CHD4 was knocked down or PADs 1 and 3 were overexpressed. As such, the authors should include in vitro assay data on mutants to validate that citrullination of these residues has the purported effects on binding of Serine or inhibitory amino acids.'

As described above for referee 1, we have now done additional experiments to address this issue. 1). We analyzed the effect of siCHD4 and ectopic expression of PADI1, PADI3 or both on PKM2 enzymatic activity rather than glycolysis in living cells. This new data shows stimulation of PKM2 enzymatic activity in each condition. Furthermore, we also analyzed PKM2 enzymatic activity in each condition in the presence of increasing quantities of Phe and Trp. We found that siCHD4 and ectopic PADI expression partially overcame the inhibitory effect of Phe and Trp. This new data shows that citrullination stimulates PKM2 enzymatic activity and that it diminishes the sensitivity of the enzyme to inhibitory amino acids. Thus, the changes in glycolysis are mirrored by the changes in activity of the PKM2 enzyme consistent with the current literature that shows the key role of PKM2 in regulation of glycolysis and its regulation by the inhibitory amino acids. This new data is shown in Figs. 3f, 5h-i and in a new supplemental Fig. 6. 2). We showed that primary WI-38 fibroblasts as expected express PKM1 and not PKM2. We then showed that glycolysis of WI-38 cells is insensitive to Phe and Trp unlike cancer

cells and that PKM1 activity is insensitive to siCHD4 silencing. Hence, sensitivity of cells to siCHD4 and presence of inhibitory amino acids correlates with the presence of PKM2. This new data is shown in supplemental Figs 5h and 6. Together these new experiments strongly point towards PKM2 as the key enzyme principally responsible for the altered glycolysis.

We made PKM2 expression vectors in which the citrullinated arginines were mutated individually or combination in lysine. This is not the ideal situation, as while arginine and lysine both have a positive charged side chain, they are not structurally equivalent. It would also have been possible to mutate the arginines to alanines, but this would have had an intrinsic effect on PKM2 structure and ligand binding even in absence of citrullination as exemplified by the R489A mutation (Morgan et al., 2013, PNAS) that abrogates FBP binding. Similarly, mutation of R106 to alanine would impact binding of Ser, Phe and Trp even in absence of citrullination. One major caveat is therefore that there are no amino acids that mimic the structural consequences of conversion of arginine to citrulline making it difficult to discriminate between intrinsic structural effects compared to loss of citrullination. Interpretation of the experiments where we transfected the vectors for the wild-type and mutated PKM2 proteins in cells is complicated by the overexpression. Citrullination leads to a subtle change in the ability of the enzyme to discriminate the activator Ser from the inhibitors Phe and Trp. PKM2 overexpression overrides the normally occurring equilibrium between the concentration of enzyme and the different cellular ligands. Properly addressing this would require reconstituting an *in vitro* reaction with the appropriate concentrations of enzyme, FBP, Ser, Phe and Trp and stoichiometric citrullination, an experiment beyond the scope of the present study. As described above, we confirmed that the effects on glycolysis seen in the living cells were matched by the changes in PKM2 enzymatic activity in cell extracts and modulation of its sensitivity to inhibition by Trp and Phe. This provides strong evidence that PKM2 activity is the key effector of the cellular glycolysis.

'5. The authors should note that Tilvawala et al (ref 41) showed that citrullination of PKM2 by PADs 1 and 3 led to an ~2-fold increase in in vitro PKM2 activity and this effect was independent of FBP binding.'

We are indeed aware of this data. In the Tilvawala study, PADI1 and 3-mediated citrullination increased PKM2 enzymatic activity in an *in vitro* reaction independent of FBP, Ser and inhibitory amino acids. We therefore cannot exclude that citrullination of one or several of the PKM2 arginines has an intrinsic stimulatory activity irrespective of the presence of activator or inhibitor ligands. We have modified the discussion to mention this point. We have also mentioned in the discussion (see above response to referee 1) that this same study showed that citrullination modulates the enzymatic activity of other glycolytic enzymes namely NNMT and ENO.

6. The authors refer to the proteins as PADI1 and PADI3. Although this annotation is used by uniprot, the proper convention is to refer to the enzymes as PADs and the genes as PADIs. This nomenclature is widely used in the citrullination field, especially the rheumatology community, where much of the citrullination literature has been centered. The authors should therefore revise their manuscript so that it follows the generally accepted format.

We take the point of the referee. We have left the Official gene symbol and Uniprot nomenclature as I believe this is what is required by Nature Communications. If the editor feels it is better to change the protein nomenclature to PAD rather than PADI, we will of course be happy to make the change.

Reviewer #3

Major points:

- More background on CHD4 mutation frequency in cancers, particularly melanoma, would add to the significance of the study. In addition, the prevalence of glycolytic metabolism in cancers with CHD4 mutation would also bolster significance of results in this manuscript.

We understand the referees comment, but as we pointed out in the original version of the manuscript the role of CHD4 in regulation of PADI1 and PADI3 expression that we observe in cancer cells in our experiments and by analyses of the cell lines in the cancer cell encyclopedia (shown in Supplemental Figs. 2 and 3) are not directly transposable to human tumours. While we do see a strong co-regulation of PADI1 and PADI3 expression in the TCGA data from many human tumours, there is no clear relationship with CHD4 expression. We were clear about this in the text and we did not wish to claim that CHD4 is the only factor involved. In human tumours, CHD4 amplification and overexpression is as prevalent as its deletion or loss of function and to our knowledge there are no data relevant to an association with glycolysis. Analyses of the TCGA datasets shows that even in uterine cancers, that display a higher frequency of CHD4 mutation and in particular truncating mutations, there is no evident correlation with PADI1 and PADI3 expression or glycolysis.

We therefore found it important to identify situations in which PADI1/PADI3 citrullination of PKM2 may be relevant for tumour biology. In the original version of the paper, we mentioned that expression of PADI1 and PADI3 were reported to be de-regulated under hypoxic conditions. We have now validated this experimentally in new data shown in Fig. 7 where we show, up-regulated PADI1 and PADI3 expression in melanoma cells grown in 3D spheroids or in hypoxic conditions without changes in CHD4 expression. Under these conditions, we observed increased PKM2 R106 citrullination and

increased PADI-dependent glycolysis. Thus, citrullination contributes to the increased glycolysis that is observed in hypoxic conditions, a hallmark of tumours in vivo.

‘ The focus on both CHD3 and CHD4 in the introduction and Fig. 1 is not clear as only CHD4 phenotypes are subsequently investigated. The manuscript may have a better logical flow if CHD3 results were deemphasized.’

We thought it was relevant to show that CHD3 and CHD4 regulate distinct gene expression programs. While, there is at least one previous report comparing the function of these two related proteins in the same cell type, these differences have not been extensively characterized which is why we compared the siCHD3 and siCHD4 results.

‘ The logic of focusing on PADI1 and 3 as critical mediators of CHD4 function is not clear. This chromatin-remodeler regulates a multitude of genes that can influence proliferation and the other phenotypes measured in this study. In order to better link PADI1/3 in these phenotypes, expression would need to be restored in siCHD4 cells.’

The referee raises an issue that we perhaps did not fully explain. Indeed, CHD4 silencing deregulates the expression of a large number of genes and as shown in Fig. 1j-k, several showed comparable de-regulated expressions in MM117 melanoma cells. Nevertheless, this is not true for many other genes. Not surprisingly, we observe cell-type specificity in the repertoire of CHD4 regulated genes. In contrast, the PADI1 and PADI3 genes are amongst the rare genes that were comparably de-regulated in all of the tested cell types an observation that correlated with the increased glycolysis. Moreover, ectopic PADI1 and PADI3 expression is both necessary and sufficient to elicit the increased glycolysis showing that these enzymes are the direct mediators of this effect, but we do not exclude that other CHD4-regulated genes could contribute to different aspects of the phenotype of the knockdown cells. We have thus modified the Figs. 1j-k to show that while some genes were comparably de-regulated in the 501Mel and MM117 cells, this is not true for all genes and we include data from a third melanoma line Sk-mel-28 that highlights the cell-specific effects of CHD4 silencing. Thus, even amongst melanoma lines, many genes show differential regulation and we have gone beyond melanoma cells to show that PADI1 and PADI3 expression and glycolysis are affected in other cancer cell types pointing to their key role in the phenotype.

‘ The mechanism of CHD4-regulated PADI1/3 expression is not entirely clear, as CHIP and RNA-seq data is limited to correlation, not causation. The authors should temper their conclusions that CHD4 prevents

activation by CTCF or other transcription factors. More experiments in siCHD4 cells would be needed to make this conclusion.'

We agree with the referee and we have modified the text to highlight that indeed our data are correlative.

- On a related note, the correlation with PADI1/3, CTCF, and CHD4 expression is interesting, but would be more informative if CHD4 mutation was also factored into these analyses. So, for example, in cancer cells with CHD4 mutation, is PADI1/3 expression altered?

As mentioned above, total CHD4 loss of function mutations in cancer are rare and it is therefore not possible to infer a direct causal relationship between CHD4 and PADI1 and PADI3 expression. There are clearly alternative regulatory pathways involved one of which is hypoxia. We have modified the text to clarify this point.

- The citrullination data in fig 2 is interesting. However, again, the focus on PKM2 is not entirely clear. Can the authors present a broader view of what pathways are enriched/depleted in siCHD4 cells? Are known substrates of PADI1/3 recovered in this assay? Moreover, a rescue experiment with restoration of PADI1/3 expression is needed to link PADI1/3 function to CHD4-mediated citrullination.

As mentioned above in response to referee 1, we have added a new supplemental figure (Fig. S4) that shows the KEGG pathway analyses of the proteins enriched in the siCHD4 samples and the overlap between citrullinated proteins enriched in the siCHD4 samples and those previously found in other studies from human tissues as well as in rheumatoid arthritis. These previous studies are cited in the revised manuscript. We do not fully understand the idea of a 'rescue' experiment. As mentioned above we show that siCHD4 enhances PADI1 and PADI3 expression with increased glycolysis. We showed that ectopic PADI1 and PADI3 expression had the same effect and that silencing of PADI1 and PADI3 in addition to that of siCHD4 abrogated the increased glycolysis. These data indicated that PADI1 and PADI3 expression is both necessary and sufficient for the increased glycolysis. We have gone on to show that CHD4 silencing as well as ectopic PADI1 and/or PADI3 expression stimulates PKM2 enzymatic activity and decreases the sensitivity of PKM2 to the effects of inhibitory amino acids.

'- In the antibody characterization in Supp fig 4, more controls are needed, such as siPADI1/3 and siPKM2.'

We have added an experiment showing that signal with the anti-R106Cit and R246Cit antibodies is lost when PKM2 is silenced and we showed that R106Cit is increased in a PADI1/3-dependent manner under hypoxic conditions.

'- For the metabolic assays in fig 3, more support is needed for the role of PADI1/3 and PKM2 citrullination in the phenotypes mediated by CHD3. More assays should be performed rescuing the siCHD4 phenotype and with PKM2 citrullination mutants.'

As mentioned above, we have included a number of additional experiments to address this issue. 1). We analyzed the effect of siCHD4 and ectopic expression of PADI1, PADI3 or both on PKM2 enzymatic activity rather than glycolysis in living cells. This new data shows stimulation of PKM2 enzymatic activity in each condition. Furthermore, we also analyzed PKM2 enzymatic activity in the presence of increasing quantities of Phe and Trp. We found that siCHD4 and ectopic PADI expression partially overcame the inhibitory effect of Phe and Trp. This new data shows that citrullination stimulates PKM2 enzymatic activity and that it diminishes the sensitivity of the enzyme to inhibitory amino acids. Thus, the changes in glycolysis are mirrored by the changes in activity of the PKM2 enzyme consistent with the current literature that shows the key role of PKM2 in regulation of glycolysis and its regulation by the inhibitory amino acids. This new data is shown in Figs. 3f, 5h-i and in a new supplemental Fig. 6. 2). We showed that primary WI-38 fibroblasts as expected express PKM1 and not PKM2. We then showed that unlike cancer cells, glycolysis of WI-38 cells is insensitive to ectopic PADI1/PADI3 expression and exogenous Phe and Trp and that PKM1 activity is insensitive to siCHD4 silencing. Hence, sensitivity of cells to citrullination and presence of inhibitory amino acids correlates with the PKM2 expression. This new data is shown in supplemental Figs 5h and 6. Together these additional experiments strongly point towards PKM2 as the key enzyme principally responsible for the altered glycolysis.

As mentioned in the response to referee 2, we made PKM2 expression vectors in which the citrullinated arginines were mutated individually or combination in lysine. This is not the ideal situation, as while arginine and lysine both have a positive charged side chain, they are not structurally equivalent. It would also have been possible to mutate the arginines to alanines, but this would have had an intrinsic effect on PKM2 structure and ligand binding even in absence of citrullination as exemplified by the R489A mutation (Morgan et al., 2013, PNAS) that abrogates FBP binding. Similarly, mutation of R106 to alanine would impact binding of Ser, Phe and Trp even in absence of citrullination. One major caveat is therefore that there are no amino acids that mimic the structural consequences of conversion of arginine to citrulline making it difficult to discriminate between intrinsic structural effects compared to loss of citrullination. Interpretation of the experiments where we transfected the

vectors for the wild-type and mutated PKM2 proteins in cells is complicated by the overexpression. Citrullination leads to a subtle change in the ability of the enzyme to discriminate the activator Ser from the inhibitors Phe and Trp. PKM2 overexpression overrides the normally occurring equilibrium between the concentration of enzyme and the different cellular ligands. Properly addressing this would require reconstituting an in vitro reaction with the appropriate concentrations of enzyme, FBP, Ser, Phe and Trp and stoichiometric citrullination, an experiment beyond the scope of the present study. As described above, we confirmed that the effects on glycolysis seen in the living cells were matched by the changes in PKM2 enzymatic activity in cell extracts and modulation of its sensitivity to inhibition by Trp and Phe. This provides strong evidence that PKM2 activity is the key effector of the cellular glycolysis.

‘ It would seem that the alteration of media supplements in fig 5 would have a broad transcriptional and metabolic effect on the cell. Can the authors elaborate on why the observed changes are influenced by PKM2 citrullination, rather than other factors?’

To address this, we performed the above described experiments showing that unlike cancer cells glycolysis in WI-38 cells is insensitive to ectopic PADI1/3 expression and exogenous Phe and Trp. Hence, sensitivity of cells to inhibitory amino acids correlates with the presence of PKM2. These observations are in full agreement with many previous observations both in cells and in vitro (cited in the original and revised versions of the manuscript) that PKM2, but not PKM1, is subject to this complex regulation by multiple ligands. Also, it is important to emphasize, that in the experiments measuring PKM2 enzymatic activity, exogenous Phe and Trp were added directly to the extracts, not to the living cells. Thus, the altered sensitivity of the enzyme to inhibition cannot result from changes in gene expression.

Minor points

‘ What is the designation of “slow proliferating cells” in Fig. 1? How is slow vs fast measured?’

We used the Invitrogen cell trace violet kit to measure cell proliferation. This assay is based on dilution of the dye that is incorporated at time 0. The graph shows the results of a typical FACS assay. T0 indicates the intensity of cells where the dye has just been added where it has not been diluted by cell division. The dye is diluted most in the siControl cells, whereas the dye is less diluted in cells transfected with siCHD3, siCHD4 and siMITF. To calculate the % slow proliferating cells, we take the cut off as the

5% or 10% slowest proliferating cells on the control and calculate the % of the cells in the other conditions that fall to the right of this value. It is also evident that compared to the siControl cells the peak of the other populations is shifted to the right indicating less dilution and hence slower proliferation.

'- For supplemental fig 3, there appear to be many possible outlier data points. Are these excluded when calculated correlation values?'

This data is taken from TCGA (cbioportal.org) and the outlier data points are taken into account in the calculated correlation values.

'- Can the authors comment on why fig 3g and k are not proportional to the level of PKM2 citrullination in fig 2?'

We do not fully understand the referee's comment. Fig. 2 measures PKM2 citrullination by immunoprecipitation followed by mass-spectrometry or immunoblot, while Fig. 3 measures glycolysis in the living cells. While our data indicate increased PKM2 citrullination in Fig. 2 and increased glycolysis in Fig. 3, it is asking too much to see a fully proportional relationship between the two assays. What is important in Fig. 3 is that we show that PADI1 and PADI3 expression is necessary and sufficient to elicit increased glycolysis.

Reviewers' comments:

Reviewer #1 (Remarks to the Author):

The revised form of the manuscript now contains new experimental data showing that PKM enzyme activity is indeed modified by citrullination in a way that is consistent with the changes measured in glycolysis. The other major question I had was about the possible role played by the constitutively active isoform PKM1 and this has also been appropriately addressed in the rebuttal. I would therefore support publication of this work.

Reviewer #2 (Remarks to the Author):

Coassolo et al (2020) Citrullination of pyruvate kinase by PADI1 and PADI3 regulates glycolysis and cancer cell proliferation

In general, the authors have responded positively to the first round of reviews, and while the results are highly interesting and will be broadly appreciated, there are several issues that remain to be addressed. These issues include:

1. The authors have now provided information on the b and y ions for the three citrullination sites they identified (i.e., R106, R246, and R489) in PKM2. Notably, for two out of the three peptides identified from this analysis, the arginine is present at the C-terminus of the peptide. Since they used trypsin and trypsin poorly cleaves after citrulline, these peptides cannot be definitively assigned as being citrullinated unless the authors can provide corroborating information including the presence of multiple neutral loss ions that result from the loss of isocyanic acid from citrulline. Note that the misassignment of residues as being citrullinated is a common bug of current software programs and is most often attributable to the incorrect assignment of the monoisotopic peak during the initial MS scan or the deamination of asparagine or glutamine. Notably both peptides that identify a C-terminal citrulline possess glutamines in their sequence. Since the citrullination of R106 and R246 was previously reported, it might be best to refer to that literature rather than present the current MS/MS data, unless of course they can detect multiple (>2) neutral loss species for each of the peptides. Please note that while several older proteomics papers have highlighted citrullines at the C-terminus of peptides, many of those studies are now suspect.

2. I apologize for using the term quantitative proteomics, however, typically when one compares sample A to sample B there is some form of quantification. Perhaps my confusion stemmed from the authors' simplistic description that 'they used program X to analyse the data.' To prevent similar confusion by other readers, it would be useful to add a few sentences (similar to those used in the response to reviewers) to clarify the methodology.

3. Although the citrullination of NNMT leads to its inactivation, NNMT is not a glycolytic enzyme. The authors should remove or edit the sentence where they discuss NNMT.

4. I would strongly urge the editor to allow the authors to use the longstanding nomenclature of the field wherein the enzymes are referred to as PADs and the genes as PADIs. This nomenclature is widely used in the citrullination field, especially the rheumatology/immunology community, where much of the citrullination literature has been centered.

Reviewer #3 (Remarks to the Author):

In the revised manuscript Coassolo et al, the authors have performed substantial work to address the reviewers' comments. However, as pointed out by this particular reviewer, the manuscript as originally presented lacked significance in the relationship between CHD4 regulation of PADI1 and 3 in tumors. The authors accurately present the lack of this relationship and again describe this in the rebuttal. As the manuscript seems to aim to uncover metabolic mechanisms in cancer cells, the lack of this connection in human cancers is concerning and suggests mechanisms relevant to in vitro cell culture only. Although the results are interesting and informative as to PADI1/3 and PKM2 function in glycolysis, in regards to cancer-relevance, the manuscript lacks significance. This

is particularly important given that the manuscript provides substantial background information on CHD4 in cancer and cancer metabolism in general. As such, the authors seem to present an expectation that is not fulfilled. In addition, as in the original version, the authors still include unrelated data and speculative models that are not addressed with additional experiments. In total, the flow of the manuscript is a bit distracting and could be improved.

Major points:

1. Regarding original reviewer's comment - More background on CHD4 mutation frequency in cancers, particularly melanoma, would add to the significance of the study. In addition, the prevalence of glycolytic metabolism in cancers with CHD4 mutation would also bolster significance of results in this manuscript.

The authors justly point out that the original manuscript describes the lack connection between CHD4 mutations and PADI1/3 expression in human cancer datasets. This is described in supplemental figures 2 and 3. However the abstract and introduction preceding these figures provide background on the importance of CHD3 and 4 in cancer. The organization in the presentation of background and results is confusing for the reader and sets up an expectation that cannot be fulfilled. The manuscript may benefit from reorganization and presentation of data that is relevant to human cancer. As currently written, it appears that some of the results might only be relevant to in vitro cell culture conditions, which is informative yet lacks substantial significance.

2. Regarding original reviewer's comment - The focus on both CHD3 and CHD4 in the introduction and Fig. 1 is not clear as only CHD4 phenotypes are subsequently investigated. The manuscript may have a better logical flow if CHD3 results were deemphasized'.

The authors do not adequately address this comment. As stated above, the authors could improve the logical flow and focus the reader's attention on relevant results by removing or de-emphasizing CHD3 introduction as this has limited investigation in the manuscript and is not relevant to the main findings.

3. Regarding the original reviewer's comment - The mechanism of CHD4-regulated PADI1/3 expression is not entirely clear, as ChIP and RNA-seq data is limited to correlation, not causation. The authors should temper their conclusions that CHD4 prevents activation by CTCF or other transcription factors. More experiments in siCHD4 cells would be needed to make this conclusion.'

The authors seem to have keep the description of CTCF-mediated PADI1/3 expression model, but have included a sentence stating that additional experiments are needed to validate. This is again distracting for the reader, as the authors provide correlative support for a model that is not validated. It is not clear that the description of this fairly lengthy speculative model is needed to support the major conclusions of the manuscript.

4. Regarding the author's response - "ectopic PADI1 and PADI3 expression is both necessary and sufficient to elicit the increased glycolysis showing that these enzymes are the direct mediators of this effect, but we do not exclude that other CHD4-regulated genes could contribute to different aspects of the phenotype of the knockdown cells. "

Although it may be in the manuscript, I could not find the sentence where the authors acknowledge that other genes may contribute to siCHD4 phenotypes. Can the authors point to this?

5. The authors should include the FACs data of proliferating cells assayed by crystal violet. Without additional information, it seems that the designation of slow versus fast proliferating cells is determined by arbitrary cutoff.

We thank the reviewers for their comments and suggestions.

REVIEWER COMMENTS

Reviewer #1 (Remarks to the Author):

The revised form of the manuscript now contains new experimental data showing that PKM enzyme activity is indeed modified by citrullination in a way that is consistent with the changes measured in glycolysis. The other major question I had was about the possible role played by the constitutively active isoform PKM1 and this has also been appropriately addressed in the rebuttal. I would therefore support publication of this work.

Reviewer #2 (Remarks to the Author):

Coassolo et al (2020) Citrullination of pyruvate kinase by PADI1 and PADI3 regulates glycolysis and cancer cell proliferation

In general, the authors have responded positively to the first round of reviews, and while the results are highly interesting and will be broadly appreciated, there are several issues that remain to be addressed. These issues include:

1. The authors have now provided information on the b and y ions for the three citrullination sites they identified (i.e., R106, R246, and R489) in PKM2. Notably, for two out of the three peptides identified from this analysis, the arginine is present at the C-terminus of the peptide. Since they used trypsin and trypsin poorly cleaves after citrulline, these peptides cannot be definitively assigned as being citrullinated unless the authors can provide corroborating information including the presence of multiple neutral loss ions that result from the loss of isocyanic acid from citrulline. Note that the misassignment of residues as being citrullinated is a common bug of current software programs and is most often attributable to the incorrect assignment of the monoisotopic peak during the initial MS scan or the deamination of asparagine or glutamine. Notably both peptides that identify a C-terminal citrulline possess glutamines in their sequence. Since the citrullination of R106 and R246 was previously reported, it might be best to refer to that literature rather than present the current MS/MS data, unless of course they can detect multiple (>2) neutral loss species for each of the peptides. Please note that while several older proteomics papers have highlighted citrullines at the C-terminus of peptides, many of those studies are now suspect.

We agree with the referee's comments. Indeed, we are aware of the caveat that trypsin is reported to cleave very inefficiently after arginine conversion to citrulline. Moreover, the technical characteristics of the mass spectrometer used for the experiments cannot exclude a miss assignment of the monoisotopic peak of the parent ions. The neutral loss signature ions are mainly detected at low intensity and cannot be used for any definitive conclusions.

We have therefore followed the referee's suggestions and modified the presentation. We have modified the text to state that; 1) we identify citrullination of R106, R246 and R279 as found in previous studies; 2) our data for citrullination of R243 and R489 are ambiguous as the residues are at the C-terminus of the peptide; 3) that citrullination of R106 and R246 was confirmed using specific antibodies, but that R489 could not be confirmed due to the insolubility of the peptides that precluded antibody generation. We modified Fig. 2 to remove the PKM2 peptides leaving only the overall PSM

counts in the two samples below the volcano plot. We added the MS/MS tables in a new Supplementary Figure 5. Note that we did not mention R279 previously, although we correctly identified this previously described residue, as no consequence of R279 citrullination on PKM2 activity could be readily predicted and hence this residue was not further studied. Furthermore, while we are explicit that we could not validate R489 citrullination, the data in Fig. 6 are consistent with this modification. This has been modified in the discussion. Nevertheless, the most critical residue in our analysis is R106 as its citrullination differentially affects the binding of the free amino acids to the regulatory pocket of PKM2, this residue was previously described and confirmed here both by mass-spectrometry and the use of specific antibodies. We have additionally modified the abstract to specifically cite R106 as the critical residue in the study.

2. I apologize for using the term quantitative proteomics, however, typically when one compares sample A to sample B there is some form of quantification. Perhaps my confusion stemmed from the authors simplistic description that 'they used program X to analyse the data.' To prevent similar confusion by other readers, it would be useful to add a few sentences (similar to those used in the response to reviewers) to clarify the methodology.

We have modified the text accordingly as suggested by the referee.

3. Although the citrullination of NNMT leads to its inactivation, NNMT is not a glycolytic enzyme. The authors should remove or edit the sentence where they discuss NNMT.

This has been removed.

4. I would strongly urge the editor to allow the authors to use the longstanding nomenclature of the field wherein the enzymes are referred to as PADs and the genes as PADIs. This nomenclature is widely used in the citrullination field, especially the rheumatology/immunology community, where much of the citrullination literature has been centered.

As suggested by the editors, we have introduced the term PAD instead of PADI at the beginning of the text and used PAD throughout the text when it refers to the protein.

Reviewer #3 (Remarks to the Author):

In the revised manuscript Coassolo et al, the authors have performed substantial work to address the reviewers' comments. However, as pointed out by this particular reviewer, the manuscript as originally presented lacked significance in the relationship between CHD4 regulation of PADI1 and 3 in tumors. The authors accurately present the lack of this relationship and again describe this in the rebuttal. As the manuscript seems to aim to uncover metabolic mechanisms in cancer cells, the lack of this connection in human cancers is concerning and suggests mechanisms relevant to in vitro cell culture only. Although the results are interesting and informative as to PADI1/3 and PKM2 function in glycolysis, in regards to cancer-relevance, the manuscript lacks significance. This is particularly important given that the manuscript provides substantial background information on CHD4 in cancer and cancer metabolism in general. As such, the authors seem to present an expectation that is not fulfilled. In addition, as in the original version, the authors still include unrelated data and speculative models that are not addressed with additional experiments. In total, the flow of the manuscript is a bit distracting and could be improved.

Major points:

1. Regarding original reviewer's comment - More background on CHD4 mutation frequency in cancers, particularly melanoma, would add to the significance of the study. In addition, the prevalence of glycolytic metabolism in cancers with CHD4 mutation would also bolster significance of results in this manuscript.

In response to the referee's concern on how PAD1 and PAD3 are regulated in tumours, we have added new data and substantially modified the text. In the first revision of the paper, we added data showing that PADI1 and PADI3 expression was up-regulated in hypoxic conditions. We have added new data showing the important role hypoxia may play in regulating PADI1 and PADI3 in human tumours. First, we used data mining in DepMap to show that PADI1 and PADI3 expression correlates with several different hypoxia signatures in 1848 cancer cell lines from the CCLE. Second, we added PADI1 and PADI3 expression data from the PanCancer Atlas showing they are expressed in a collection of tumours known to be associated with hypoxia. Third, we generated a hypoxic score for each patient sample in several cancers notably pancreatic adenocarcinoma, clear cell renal carcinoma and lung adenocarcinoma and found that PADI1 and PADI3 positively correlated with the hypoxia score. This data has been added as a new Supplementary Fig. 10 and the correlation with patient samples has been included in Fig. 7. We have reorganized the text to clearly state that; 1) while CHD4 negatively regulates PADI1 and PADI3 expression in cell lines this does not seem to be the case in human tumours hence there may be other regulatory mechanisms; 2) that PADI1 and PADI3 expression is positively correlated with hypoxia, both in the CCLE and in several human tumours; 3) these observations taken together with the experimental results that we included in the previous version reinforce the idea that PKM2 citrullination contributes to stimulating glycolysis in hypoxic conditions, a hallmark of many solid tumours and rheumatoid arthritis tissue.

To specifically answer the referees comment; CHD4 is not frequently mutated in cancer and we know of no data sets where we can assess a relationship between CHD4 mutation and the metabolic status of the tumour. We are not aware of any data set that comprehensively describes the metabolic status of human tumours. We could mine the TCGA data for a signature of altered expression of glycolytic enzymes, however this is not relevant as PAD1 and PAD3 do not act at the gene expression level, but rather directly on PKM2 enzymatic activity. We rather propose that the association of PAD1 and PAD3 expression with hypoxia gives a much better model for the implication of PKM2 citrullination in metabolism both in cell lines and in tumours.

The authors justly point out that the original manuscript describes the lack connection between CHD4 mutations and PADI1/3 expression in human cancer datasets. This is described in supplemental figures 2 and 3. However the abstract and introduction preceding these figures provide background on the importance of CHD3 and 4 in cancer. The organization in the presentation of background and results is confusing for the reader and sets up an expectation that cannot be fulfilled. The manuscript may benefit from reorganization and presentation of data that is relevant to human cancer. As currently written, it appears that some of the results might only be relevant to in vitro cell culture conditions, which is informative yet lacks substantial significance.

As mentioned above, we have substantially modified the text as suggested by the referee. We have removed most of the data and references to CHD3. We kept only the data necessary to show the silencing of CHD4 was specific with no cross reaction with CHD3. We removed all of the data on CTCF and FOS/Jun from the Supplementary Figures leaving only the data on the correlated expression of PADI1 and PADI3 and the anti-correlation with CHD4 in the CCLE. We removed all reference to the role

of CTCF and AP1 from the text. As described above, we reorganized the text to highlight the new data on the role of hypoxia.

2. Regarding original reviewer's comment – The focus on both CHD3 and CHD4 in the introduction and Fig. 1 is not clear as only CHD4 phenotypes are subsequently investigated. The manuscript may have a better logical flow if CHD3 results were deemphasized'. The authors do not adequately address this comment. As stated above, the authors could improve the logical flow and focus the reader's attention on relevant results by removing or de-emphasizing CHD3 introduction as this has limited investigation in the manuscript and is not relevant to the main findings.

As mentioned above we removed almost all reference to CHD3 from the text including the RNA-seq data. The text focusses now only on CHD4.

3. Regarding the original reviewer's comment - The mechanism of CHD4-regulated PADI1/3 expression is not entirely clear, as ChIP and RNA-seq data is limited to correlation, not causation. The authors should temper their conclusions that CHD4 prevents activation by CTCF or other transcription factors. More experiments in siCHD4 cells would be needed to make this conclusion.'

The authors seem to have keep the description of CTCF-mediated PADI1/3 expression model, but have included a sentence stating that additional experiments are needed to validate. This is again distracting for the reader, as the authors provide correlative support for a model that is not validated. It is not clear that the description of this fairly lengthy speculative model is needed to support the major conclusions of the manuscript.

We agree with the referee and we have removed the correlation data and the mention of this mechanism involving CTCF and AP1 from the revised text.

4. Regarding the author's response – "ectopic PADI1 and PADI3 expression is both necessary and sufficient to elicit the increased glycolysis showing that these enzymes are the direct mediators of this effect, but we do not exclude that other CHD4-regulated genes could contribute to different aspects of the phenotype of the knockdown cells. " Although it may be in the manuscript, I could not find the sentence where the authors acknowledge that other genes may contribute to siCHD4 phenotypes. Can the authors point to this?

This sentence was on bottom of page 15 and top of 16 in the previous version and now page 16 of the new version.

5. The authors should include the FACs data of proliferating cells assayed by crystal violet. Without additional information, it seems that the designation of slow versus fast proliferating cells is determined by arbitrary cutoff.

We have made a new Supplementary Figure (Fig. S1) to show representative FACS assays illustrating how the method is used. While the cutoff can be set in various ways, the clear shift in the FACS profile towards to right indicates a global shift in the dilution of the dye and hence proliferation of the cell populations. T0 shows the reference cells that have not divided and hence the shift towards this value is indicative of slower proliferation. This can be clearly observed in the examples we show in the new Supplemental Figure.

REVIEWERS' COMMENTS

Reviewer #2 (Remarks to the Author):

The authors have fully addressed my concerns. I did identify one minor error that the authors should correct. Specifically, on page 10 they note that 'the carboxyl groups of V215 and L217 interact...'. I believe carboxyl groups should be 'main chain carbonyls'. This correction does not require rereview. I would also note that the authors should be commended for their very positive responses to the prior reviews and the exciting data presented in the manuscript.

Reviewer #3 (Remarks to the Author):

The authors have addressed all the previous comments. I now believe it is suitable for publication.

We are delighted that the referees are now all in favour of publication. We have corrected the small typo 'carboxyl' changed to carbonyl' as pointed out by referee. We appreciate the thoughtful comments of referee 2 in the last round of revision.